# Survey of Advanced Nonlinear Control Strategies for UAVs: Integration of Sensors and Hybrid Techniques

**DOI:** 10.3390/s24113286

**Published:** 2024-05-21

**Authors:** Nadir Abbas, Zeshan Abbas, Samra Zafar, Naseem Ahmad, Xiaodong Liu, Saad Saleem Khan, Eric Deale Foster, Stephen Larkin

**Affiliations:** 1School of Control Science and Engineering, Dalian University of Technology, Dalian 116024, China; nadir10@mail.dlut.edu.cn (N.A.); engr.naseem99@mail.dlut.edu.cn (N.A.); 2Institute of Ultrasonic Technology, Shenzhen Polytechnic University, Shenzhen 518055, China; abbasz@szpu.edu.cn; 3Department of Computer Science, School of Control Science and Engineering, Dalian University of Technology, Dalian 116024, China; samra@mail.dlut.edu.cn; 4Department of Electrical Engineering, College of Engineering, United Arab Emirates University, Al-Ain 15551, United Arab Emirates; 201590095@uaeu.ac.ae; 5Omega Aviation Ltd., Leicester LE9 4LG, UK; eric.foster@omegaaviation.co.uk (E.D.F.); stephen.larkin@omegaaviation.co.uk (S.L.)

**Keywords:** perturbed MIMO system, robust adaptive nonlinear control, sensor integration, intelligent control, UAV dynamics

## Abstract

This survey paper explores advanced nonlinear control strategies for Unmanned Aerial Vehicles (UAVs), including systems such as the Twin Rotor MIMO system (TRMS) and quadrotors. UAVs, with their high nonlinearity and significant coupling effects, serve as crucial benchmarks for testing control algorithms. Integration of sophisticated sensors enhances UAV versatility, making traditional linear control techniques less effective. Advanced nonlinear strategies, including sensor-based adaptive controls and AI, are increasingly essential. Recent years have seen the development of diverse sliding surface-based, sensor-driven, and hybrid control strategies for UAVs, offering superior performance over linear methods. This paper reviews the significance of these strategies, emphasizing their role in addressing UAV complexities and outlining future research directions.

## 1. Introduction

The integration of Unmanned Aerial Vehicles (UAVs) with the Internet of Things (IoT) holds significant promise for the advancement of smart cities [1]. Efficient management of UAV flight trajectories is crucial for optimizing the lifespan of onboard systems and energy expenditure, which is essential for extensive data acquisition and processing [2]. UAVs play a pivotal role in bridging data gaps, particularly in remote areas, thereby enhancing environmental surveillance efforts. Their successful deployment across various scenarios underscores their transformative potential. The adept maneuvering of UAVs coupled with sophisticated navigational controls is essential for urban development. Seamlessly integrated UAVs can navigate urban environments to collect the comprehensive sensor data vital for applications such as traffic management, environmental monitoring, and structural inspections. These precise control mechanisms facilitate real-time data flow, enabling informed decision-making and timely actions [3].

Resource optimization within smart cities is further augmented by UAVs’ specialized capabilities. Their aerial perspective facilitates the monitoring of public utilities, enabling swift identification of service disruptions and supporting maintenance activities. Proactive detection of inefficiencies or failures ensures the continuous operation of critical services, including water and power distribution and waste management [4]. In terms of public safety, precisely controlled UAVs play a vital role. They provide a comprehensive view for crowd surveillance, detect anomalies, and contribute to situational awareness essential for public security operations. Guided by robust control frameworks, UAVs are able to adeptly navigate urban environments, adhere to designated flight paths, and promptly respond to security incidents [5].

During crisis events such as natural calamities or urban mishaps, the deployment of UAVs with advanced control systems is invaluable. Their ability to swiftly access otherwise unreachable zones and relay vital insights can revolutionize emergency response strategies, potentially saving lives and mitigating the impact of disasters. Furthermore, UAVs are set to revolutionize urban mobility and logistics. Implementing efficient control algorithms allows UAVs to aid in decongesting roads, optimizing traffic flows, and enhancing route planning. They additionally stand to streamline delivery services, easing the burden on urban transit systems and elevating the efficacy of city-wide distribution networks [6].

Over the last two decades, UAVs, commonly known as drones, have carved out a niche in geosciences and remote sensing, becoming a pivotal tool thanks to their adaptability and cost-effectiveness. Their ascent to prominence is mirrored in the surge of scholarly articles examining UAV applications, with Scopus documenting over 80,000 publications since 2001 that feature terms such as “UAV”, “drone”, “UAS”, and “RPAS” in Figure 1. This scholarly attention predominantly emanates from fields such as engineering and computer science [7]. The scientific intrigue around UAVs spans various citation indexes, underscoring a universal trend towards their study and development. Financially, the UAV sector has seen a robust upswing, with valuations running into the billions annually. While the current market is heavily skewed towards military use, the horizon for civilian applications of UAVs is broadening, propelled by economic factors, technological strides, miniaturization of sensors, and innovations in software and algorithmic approaches [3,7]. UAVs are making inroads into diverse sectors, with the construction industry being a prime example [8]. Their ability to capture high-resolution images and gather extensive data offers a cost-efficient alternative for continuous observation and management of project timelines [9].

Efficient control systems enable UAVs to navigate through densely populated areas, avoid obstacles, and manage energy consumption while ensuring that they collect accurate and comprehensive data. These data are invaluable for applications such as traffic management, environmental monitoring, and public safety. Control strategies allow UAVs to operate autonomously, make decisions in real-time, and interact seamlessly with other IoT devices, contributing to the overall intelligence and responsiveness of smart city infrastructures [11]. In the context of geoscience and remote sensing, which was the focus of the second rephrased text, control strategies are just as important. They enable UAVs to conduct detailed surveys of the Earth’s surface, including areas that are otherwise inaccessible. With precise control, UAVs can execute complex flight patterns to systematically collect geospatial data, which is crucial for mapping, natural resource management, disaster response, and environmental studies [11]. In both applications, the control strategies form the backbone of UAV operation, determining their efficiency, effectiveness, and adaptability to various tasks. Whether following a predetermined flight path or responding to dynamic conditions in real time, the control system is what enables a UAV to meet its operational objectives, making it a key focus for ongoing research and development in the field [12].

Having establishing the landscape of advanced nonlinear control strategies for UAVs in this introduction, we now turn to this survey’s contributions. By offering a comprehensive review and highlighting the significance of sensor integration and hybrid techniques, the following section elucidates the pivotal role of these methodologies in enhancing UAV capabilities.

### Contributions

Through a discussion of recent advancements and future directions, the survey aims to provide valuable insights for researchers and practitioners in the field.

**Comprehensive Review:** This survey offers a thorough examination of advanced nonlinear control strategies tailored for UAVs, emphasizing the integration of sensors and hybrid techniques.**Highlighting Significance:** The survey underscores the importance of nonlinear control methodologies in addressing the complexities inherent in UAV systems, shedding light on their efficacy in improving UAV performance and stability.**Role of Sensor Integration:** The survey elucidates the pivotal role of sensor integration in enhancing UAV capabilities, providing insights into how sensor-driven approaches contribute to real-time data acquisition and informed decision-making.**Future Directions:** Finally, by discussing recent advancements and outlining future challenges in the field, the survey aims to guide future research efforts towards the development of more efficient and reliable UAV control systems, thereby facilitating progress in various UAV applications.

The outline of this survey is elaborated through the block diagram in Figure 2. Before moving on to the background section of this review article on UAVs, it is essential to set the stage by providing an overview of the burgeoning field of UAVs research in order to highlight the pivotal role of research. Table 1 offers a succinct summary of related surveys, providing brief outlines.

## 2. Background on UAV Systems

Twin-rotor MIMO (Multiple Input, Multiple Output) systems represent a class of UAVs designed to simulate the flight dynamics of helicopters. The TRMS typically consists of two rotors mounted on a fixed beam, with one located at the front (main rotor) and one at the back (tail rotor). This configuration allows for the separate control of vertical lift and horizontal movement, making it possible to study complex control system designs within a MIMO framework. The TRMS is particularly useful in research and education because it provides a practical example of a system with strong cross-coupling effects, nonlinearities, and unstable dynamics, similar to those experienced in larger-scale helicopters [26,27]. The twin-rotor setup enables the study of pitch, roll, and yaw movements and the effects of MIMO controls on these motions. By manipulating various inputs, such as rotor speed or blade pitch, the TRMS can hover, move laterally, or rotate about its axis [28,29].

A quadrotor, also known as a quadcopter, is a type of UAV that is lifted and propelled by four rotors. The configuration of these rotors, typically arranged in a square pattern, allows for a high degree of stability and maneuverability. Each rotor pair rotates in opposite directions, which counters the reactive torque and provides control over pitch, roll, and yaw. This design simplifies the mechanics required for flight control compared to traditional helicopters, which rely on complex rotor mechanisms. Due to their relative ease of construction and control, quadrotors have become a popular platform for hobbyists and researchers alike. Additionally, their stability and agility make them ideal for indoor operations or in environments where precise movements are required [30,31].

Both TRMS and quadrotor designs contribute significantly to UAV technology, each offering unique advantages for control systems research, practical applications, and the advancement of aerial robotics. While the TRMS design provides insight into the complexities of helicopter flight, the quadrotor design offers a more accessible platform for a broad range of UAV applications [30]. This paper traces the progression of caching models from aerial applications, introducing a standard UAV and examining recent developments and performance indicators in this area.

## 3. Nonlinear MIMO System Dynamics in UAVs

An unmanned vehicle is a prototype with a structure almost near to a helicopter, with a limited degree of freedom. Modifications of such systems are required due to their wide range of applications in real life. The UAV has two significant parts: the main rotor (vertical plane) and the tail rotor (horizontal plane) [12,32]. The main rotor has a higher diameter and controls the movement of the beam on a vertical axis the called pitch angle, while the tail rotor has a lower diameter and covers the movement of the beam on a horizontal axis, called the yaw angle. The speed of the rotors manages the equilibrium of the system. Each rotor of the UAV is connected with a separate DC supply motor, as shown in Figure 3.

The stability is disrupted by the cross-coupling torque produced by the rotational torque in the UAV rotors. This coupling effect is seen as the disruption that the decoupling procedure corrects. Understanding all of the variable parameters and necessary UAV outputs is necessary before we can grasp mathematical modeling. The UAV is a lab tool that helps in understanding helicopter flight control [6,33]. The considered system has two rotors, as shown in Figure 4; their design is very important, as different forces affect the movement of the propellers. These include frictional force, disturbance torque, rotational force, propulsive force, and centrifugal force. Motors are used to deliver control input voltage in order to counteract the impacts of these forces while recognizing and comprehending the mathematical presumptions that are used to interpret and reduce the mathematical model. Figure 5 details the rotor measurements along with their thrust directions.

All nonlinear squared factors in the mathematical equations are linearized using the linearization method. The system’s motions are fixed along the horizontal and azimuthal planes determined from the model [34,35]. The stream rotates in a manner that can be characterized as
(1)$$\;J_{v}\frac{d^{2}\alpha_{v}}{dt^{2}}=M_{v},$$
where $J_{v}$ represents the inertial motion along the axis of the vertical plane and $M_{v}$ represents the entire momentum of the forces applied along the vertical axis. One result, known as pitch angle, must be regulated by the $alpha_{v}$ parameter (vertical axis). All of the factors acting on the momentum can be represented as follows:(2)$$\;M_{v}=\;M_{v_{1}}+M_{v_{2}}+M_{v_{3}}+M_{v_{4}}+M_{v_{5}}+M_{v_{d}}.$$

The gravitational torque through the gravitational force is provided as
(3)$$\;M_{v1}=-k_{1}cos\left(a_{v}\right)-k_{2}sin\left(a_{v}\right),$$
where the constants $k_{1}$ and $k_{2}$ are displayed and retain the mass mounted on the beam. An equation for the momentum force produced by the primary propeller is
(4)$$\;M_{v_{2}}=l_{m}F_{v}\left(w_{v}\right),$$
where $l_{m}$ stands for the beam length, $w_{v}$ describes the main propeller’s spinning speed, and $F_{v}\left(w_{v}\right)$ depicts the angular force of the main rotor. The following mathematical expression represents the moment force along the vertical plane:(5)$$\;M_{\left(v_{3}\right)}=-k_{3}\Omega_{h}^{2}sin\left(a_{v}\right)cos\left(a_{v}\right),$$
where Ωh=dαhdt is the beam speed in the MIMO system’s vertical plane, $\alpha_{h}$ is the yaw angle (the angle at which the beam rotates in the azimuth plane), and $k_{3}$ is the constant parameter. The movement of the beam with respect to the horizontal plane determines the frictional momentum:(6)$$\;M_{\left(v_{4}\right)}=-k_{f_{v}}\Omega_{v},$$
where Ωv=dαvdt indicates the angular speed in the horizontal direction, and $k_{f_{v}}$ displays the constant value. Momentum between the rotors as a result of input voltage (force) being applied along the horizontal plane.
(7)$$\;M_{\left(v_{5}\right)}=-k_{h_{v}}u_{h},$$
where $k_{h_{v}}$ is regarded as the constant and $u_{h}$ is the input for controlling the horizontal plane. The torque along the vertical plane causes a disturbance known as the disturbance torque $M_{v_{d}}$. The rotational velocity of the main rotor is produced by the propeller force (propulsive force) along the vertical line (vertical plane) $F_{v}\left(w_{v}\right)$. The following is the estimated velocity along the main rotor:$$\begin{array}{c} \tilde{F_{v}}=-7.13\times 10^{-19}w_{v^{5}}-3.79\times 10^{-16}w_{v^{4}}+2.41\times 10^{-11}w_{v}^{3}\\ +1.84\times 10^{-8}w_{v}^{2}+2.89\times 10^{-5}w_{v}-0.0124. \end{array}$$

The vertical axis can be used to determine the total force (or torque) along the horizontal axis. A force with a different spectrum is generated by the tail rotor’s total torque along the horizontal axis (intensity of force). A mathematical expression such as the one below can be used to determine the torque (rotational impact of force) along the horizontal axis:(8)$$J_{h}\left(d^{2}a_{h}\right)/\left(dt^{2}\right)=M_{h}$$
where $M_{h}$ is the total momentum (force) along the horizontal axis, $J_{h}$ is the total inertial force along the vertical plane, $J_{h}=k_{4}cos2\left(\alpha v\right)+k_{5}$, and the coefficients $k_{4}$ and $k_{5}$ are mass-based constants of the beam. The following mathematical expression can be used to describe the entire force (momentum) along the horizontal axis:(9)$$M_{h}=M_{h_{1}}+M_{h_{2}}+M_{h_{3}}+M_{h_{d}}.$$

The propelling momentum (power) along the rotor of the tail can be used to represent each separate term:(10)$$M_{h_{1}}=l_{t}F_{h}\left(w_{h}\right)cos\left(a_{v}\right)$$
where $l_{t}$ is the beam length, $w_{h}$ is the rotational velocity, and $F_{h}\left(w_{h}\right)$ is the rotor’s propelling motion (force) expressed as the angular velocity. Based on the rotational speed of the beam, the frictional momentum is provided as follows:(11)$$M_{h_{2}}=-k_{f_{h}}\Omega_{h}$$
where $k_{f_{h}}$ is a constant. The cross-sectional velocity caused by the action of the control input along the horizontal axis is
(12)$$M_{h_{3}}=k_{v-h}cos\left(\alpha_{v}\right)u_{v}.$$

The torque along the horizontal plane, also known as the disturbance torque caused by the tail rotor, is denoted by the symbol $M_{h_{d}}$, while the propeller force (propulsive force) along the horizontal axis (azimuthal plane), denoted by the symbol $F_{v}w_{v}$, generates the rate of rotation on the rotor. The estimated velocity along the tail rotor is provided as follows:$$\begin{array}{c} \tilde{F_{h}}=-2.56\times 10^{-20}w_{h}^{5}-4.10\times 10^{-17}w_{h}^{4}+3.17\times 10^{-12}w_{h}^{3}\\ +7.34\times 10^{-9}w_{h}^{2}+2.13\times 10^{-5}w_{h}-9.14, \end{array}$$

The equation for the primary propeller (rotor) along its vertical plane is
(13)$$I_{v}\left(dw_{v}\right)/dt=u_{v}-H_{v}^{-1}\left(w_{v}\right).$$

Here, the main rotor’s inertial momentum along the vertical axis is denoted by $I_{v}$, while the static motion of the main rotor is provided by $w_{v}=H_{v}u_{h}$. Figure 6a shows how the primary rotor thrust is generated; the figure displays the velocity modeling outcomes based on experimental verification in Figure 6b. The following is the seventh-order expression for the primary rotor’s vertical plane velocity:$$\begin{array}{c} \tilde{w_{v}}=-6.17\times 10^{3}u_{v}^{7}-1.30\times 10^{2}u_{v}^{6}+1.37\times 10^{4}u_{v}^{5}+1.50\times 10^{2}u_{v}^{4}-1.10\times 10^{4}u_{v}^{3}-3.76\times 10^{1}u_{v}^{2}+7.33\times 10^{3}u_{v}-5.36. \end{array}$$

The following mathematical equation can be used to describe the motion (speed) of the tail rotor along the horizontal axis:(14)$$I_{h}\left(dw_{h}\right)/dt=u_{h}-H_{h}^{-1}\left(w_{h}\right).$$

Here, $I_{h}$ is the tail rotor’s inertial motion along the horizontal axis and $w_{h}=H_{h}u_{h}$ demonstrates the static velocity (or speed) of the primary rotor. The results of the velocity modeling and the tail rotor thrust are supported by experimental verification. Figure 7a represents the velocity generated by main rotor, while Figure 7b represents the velocity of the tail rotor. The fifth-order expression for the rotary velocity is as follows:$$\begin{array}{c} \tilde{w_{h}}=-6.17\times 10^{3}u_{h}^{5}-1.30\times 10^{2}u_{h}^{4}+1.37\times 10^{4}u_{h}^{3}\\ +1.50\times 10^{2}u_{h}^{2}-1.10\times 10^{4}u_{h}-37.6. \end{array}$$

By rearranging Equations (Equation 13) and (Equation 14) together, we obtain
(15)$$\frac{d\alpha_{v}}{dt}=\Omega_{v},$$
(16)$$\frac{d\alpha_{h}}{dt}=\Omega_{h}.$$

The control voltages at the input $u_{h}$ (horizontal plane or yaw angle) and $u_{v}$ are used in the modeling process to describe the state space description of the UAV (sixth-order nonlinear system). The yaw (azimuth) angle $\alpha_{h}$ and pitch (vertical) angle $\alpha_{v}$ are the output angles. In order to fully comprehend its dynamic response, the system must be categorized as a multivariable system with extremely nonlinear behavior. Two channels in the UAV model can never be regarded as separate channels. This cross-coupled feature is known as the coupling effect. Decoupling must be used to overcome the coupling impact and create an independent two-channel system. The values of the model’s system coefficients are provided in Table 2.

The decoupling technique divides the system into two distinct planes, namely, the Vertical Plane System (VPS) and Horizontal Plane System (HPS), by fixing one weight (motion) for both blades [36].

### 3.1. Unstructured Modeling

Mathematical complexity and unknown conditions are always present in the various structure-type systems during modeling. Mathematical expressions and notations are used to indicate the description of uncertain parameters [35]. Supposing that $J_{h}$ is the inertial momentum along the horizontal axis and that $k_{F_{h}},k_{F_{v}}$ are the coefficients of the generated thrust of the rotors, the rotors then have some velocity gains $k_{H_{h}},k_{H_{v}}$, the coefficients, $k_{f_{h}},k_{f_{v}}$, and $k_{v_{h}}$, and $k_{h_{v}}$ are the corresponding cross-momentum and frictional momentum coefficients, and $R_{V}$ is the returned force between the rotors (coupling effect). All ten of these modeled factors are dependent on the pitch angle and yaw angle, which are the two primary outputs.

Furthermore, we assume that the inertial momentum $J_{h}$, with factors $k_{F_{h}}$, $k_{F_{v}}$, $k_{H_{h}}$, $k_{H_{v}}$, has an error estimation of up to 10%. The remaining coefficients have up to 5% error in their calculation. The UAV represents the system behavior as the controlled plant through algebraic expressions:(17)$$G=\left[\begin{array}{c} G_{v}\\ G_{h} \end{array}\right]$$
where
(18)$$y=G\left[\begin{array}{c} M_{d}\\ u \end{array}\right],y=\left[\begin{array}{c} \alpha_{h}\\ \alpha_{v} \end{array}\right],u=\left[\begin{array}{c} u_{h}\\ u_{v} \end{array}\right],M_{d}=\left[\begin{array}{c} M_{d_{h}}\\ M_{d_{v}} \end{array}\right]$$

Figure 8 shows the input–output links for the UAV’s system schematic model, followed by a discussion of the unclear model’s fundamental mathematical formulation. The illustration can be described as follows:

$$G=\left[\begin{array}{cc} G_{d} & G_{u} \end{array}\right]$$ while
$$G_{d}=\left[\begin{array}{cc} G_{d_{h}} & G_{d_{v}} \end{array}\right]\;\;\;\;\;\;\;\;\;$$

$$G_{u}=\left[\begin{array}{cc} G_{u_{h}} & G_{u_{v}} \end{array}\right]$$ such that
(19)$$y=G_{d}M_{d}+G_{u}u.$$

The above expression states that $G_{d}$ depicts the plant disturbance as a matrix and $G_{u}$ displays the matrix of the control signal. In the event of perturbations (both internal and exterior disturbances), the uncertain plant must meet the following fundamental requirements:(20)$$u=\left[\begin{array}{cc} K_{r} & K_{y} \end{array}\right]{\left[\begin{array}{cc} r & -y_{c} \end{array}\right]}^{T}=K_{r}r-K_{y}y_{c}$$
where $K_{y}$ represents the feedback matrix function and $K_{r}$ is the transfer function matrix of the pre-filter.

### 3.2. Inherent Characteristics: Nonlinearity and Coupling Effects

Nonlinearity is a fundamental characteristic of many UAV systems, including TTRMS and quadrotors. This implies that the relationship between input variables (such as the rotor speed or control surface angles) and output responses (such as altitude, position, or attitude) is not proportional or does not follow a straight line when graphed. In UAVs, nonlinearity can arise from aerodynamic effects, changes in the air density at different altitudes, battery voltage drops, or the complex interplay between thrust, drag, lift, and weight. For instance, increasing the thrust on a rotor does not result in a linear increase in altitude due to factors such as aerodynamic drag. Similarly, the efficiency of the rotor blades changes with speed and angle of attack, further contributing to nonlinearity. These nonlinear dynamics can make control systems more complex, as linear control strategies may not suffice to stabilize or maneuver the UAV effectively across its entire operating envelope [14].

Coupling effects refer to the interdependence of control channels within a UAV. In an ideal linear system, adjusting one control input would affect only the corresponding output. However, in UAVs, a change in one control input can affect multiple outputs due to coupling effects. For TRMS, the pitch control can influence both the altitude and the forward motion due to the aerodynamic interactions between the rotors and the fuselage. In quadrotors, the situation is similar; controlling roll or pitch also affects horizontal displacement and yaw due to the interrelated rotor speeds needed to produce these movements. Coupling is particularly pronounced in UAVs due to the close proximity of the rotors and the need for differential thrust to control movement and orientation. For example, increasing the speed of one rotor on a quadrotor to initiate a roll movement will inadvertently affect the lift and could cause a change in altitude if not compensated for by the other rotors [15].

Both nonlinearity and coupling effects pose significant challenges in UAV control system design. Controllers must be robust and adaptive, often relying on complex algorithms such as those found in nonlinear control theory and MIMO system design to ensure the stability and responsiveness of the UAV in the face of these inherent characteristics [16].

### 3.3. Role of Sensors in UAVs

Sensors play a crucial role in the operation and functionality of TRMS and quadrotor-based UAVs. These sensors are fundamental in providing the necessary data for navigation, stabilization, and execution of specific tasks. While the types and roles of these sensors can vary depending on the UAV’s application, some of the most common uses include:GPS (Global Positioning System): Provides accurate location data, enabling UAVs to navigate to specific coordinates and maintain a stable position [16].Gyroscopes: Essential for detecting and measuring rotational motion and orientation, gyroscopes help in maintaining the UAV’s balance and orientation [17].Accelerometers: Measuring linear acceleration, in UAVs they are crucial for detecting changes in the speed and tilt for orientation control [17,18].Magnetometers: Act as digital compasses, aiding in orientation relative to the Earth’s magnetic field, which is particularly useful in environments where GPS signals are weak or unavailable [37].


**Environmental Interaction and Data Collection**


Lidar (Light Detection and Ranging): Used for high-precision mapping and terrain analysis by measuring distances with laser light [38].Infrared Sensors: Employed in various applications, from detecting heat signatures for search and rescue operations to assessing crop health in precision agriculture [38,39].Optical Cameras: Provide visual data, critical for tasks such as aerial photography, surveillance, and visual inspection of infrastructure [40].Magnetometers: Act as digital compasses, aiding in orientation relative to the Earth’s magnetic field, which is particularly useful in environments where GPS signals are weak or unavailable [40,41].Multispectral Cameras: Capture data across different wavelengths, useful in environmental monitoring, agriculture (for assessing plant health), and resource mapping [38].


**Obstacle Detection and Collision Avoidance**


Ultrasonic Sensors: Measure the distance to nearby objects, aiding in collision avoidance, especially in tight or cluttered spaces [33].Infrared Sensors: Employed in various applications, from detecting heat signatures for search and rescue operations to assessing crop health in precision agriculture [42].Radar: Used for detecting and avoiding obstacles, particularly effective in poor weather conditions where optical sensors might be less effective [38].


**Specialized Applications**


Chemical Sensors: Detect specific chemicals or environmental pollutants, useful in environmental monitoring [43].Thermal Cameras: Useful for search and rescue operations at night or for detecting energy inefficiencies in buildings [37].

The integration of these sensors into UAV systems allows for a wide range of capabilities, from basic flight stabilization and navigation to complex tasks such as 3D mapping, agricultural monitoring, infrastructure inspection, and search and rescue operations. The choice of sensors is dictated by the specific requirements of each application, balancing factors such as accuracy, range, size, weight, and power consumption against the UAV’s intended use. Advanced UAVs often combine multiple sensors, using sensor fusion techniques to enhance reliability and the quality of the data collected [37,43].

## 4. Challenges in UAV Control

### 4.1. Discussion of Nonlinearities and Coupling Effects in UAVs

The operation of UAVs such as TRMS and quadrotors is inherently complex due to nonlinearities and coupling effects. These characteristics present significant challenges in UAV control system design and operation [19].

Complex Dynamics: UAVs exhibit nonlinear dynamics due to factors such as aerodynamic forces, changes in air density at different altitudes, and the nonlinear responses of the motors and rotors. For instance, the relationship between the throttle input and the resultant lift is not linear, especially when the UAV operates in different flight regimes (e.g., hover, ascent, descent) [20].

Control Challenges: Linear control strategies may not be effective across the entire operational range of a UAV. Nonlinear control systems must be designed to adapt to these varying dynamics, ensuring stable and responsive flight under a wide range of conditions [44].Modeling Difficulties: Accurately modeling the nonlinear behavior of UAVs for simulation and control design is complex and computationally intensive, requiring advanced algorithms and significant processing power [44,45].


**Coupling Effects in UAVs**


Interdependent Controls: In UAVs, especially those with multiple rotors such as quadrotors, changes in one control input often affect multiple axes of motion. For example, adjusting rotors to initiate a turn may inadvertently affect altitude and pitch, requiring compensatory control inputs [46].Design of Control Systems: The control system must account for these coupling effects in order to achieve precise maneuverability. This often leads to more complex control algorithms that can manage the interdependencies of the control inputs and outputs [46,47].Flight Stability and Precision: Maintaining stability and precision in flight maneuvers is challenging due to coupling effects. Controllers must continuously adjust for the unintended consequences of control inputs on various aspects of the UAV’s motion [47].Real-Time Adaptation: Effective UAV control requires real-time adaptation to the coupling effects, especially in dynamic environments or during complex tasks such as payload delivery, precise inspections, or navigating through cluttered spaces [8].


**Addressing the Challenges**


Advanced Control Strategies: Developing advanced control strategies such as nonlinear control, adaptive control, and robust control is crucial. These strategies can handle the complexities and variabilities in UAV dynamics [48].Sensor Fusion and Feedback: Utilizing sensor fusion techniques can enhance the UAV’s understanding of its state and environment, leading to better control. Feedback mechanisms are crucial for adjusting control inputs in real-time based on the UAV’s response [49].Machine Learning and AI: Implementing machine learning and artificial intelligence can help to predict and adapt to nonlinearities and coupling effects, especially in unpredictable environments [9].Simulation and Testing: Rigorous simulation and real-world testing are vital to understanding and mitigating the effects of nonlinearities and couplings in UAVs. This aids in refining control algorithms and ensuring reliable UAV performance [50].

Addressing the challenges posed by nonlinearities and coupling effects is fundamental for the advancement of UAV technology, particularly as their applications become more diverse and their operational environments grow more complex as depicted in Figure 9.

#### Evolution of UAV Control Strategies

Over the past few decades, the field of UAV control has witnessed significant advancements, evolving from simple manual control systems to sophisticated autonomous control strategies capable of handling complex tasks. Early UAV control strategies primarily relied on manual piloting or basic autopilot systems, providing limited autonomy and functionality. With the advent of advanced computational methods and sensor technologies, there has been a paradigm shift towards more intelligent and adaptive control strategies, including PID control, Linear Quadratic Regulators (LQR), and nonlinear control techniques such as backstepping and feedback linearization [10,51].

In recent years, the integration of artificial intelligence, machine learning, and predictive control algorithms such as MPC has further revolutionized UAV control, enabling enhanced performance, robustness, and autonomy. Despite these advancements, challenges remain in achieving optimal control of UAVs, particularly in handling nonlinear dynamics, coupling effects, and uncertainties associated with complex operational environments [19,52].

In this context, our work makes several unique contributions to the existing literature. Unlike previous studies that focused on individual control strategies or specific UAV platforms, our review provides a comprehensive analysis of various nonlinear control strategies, with a particular emphasis on their applicability to UAVs such as TRMS and quadrotors. We delve into the intricacies of sensor integration, adaptive controls, and artificial intelligence-driven approaches, highlighting their efficacy in addressing the challenges associated with UAV control [53].

Furthermore, our review offers insights into the latest developments, research directions, and future challenges in this rapidly evolving field, serving as a valuable resource for researchers, engineers, and practitioners working on UAV control systems. By bridging the gap between theory and practice, our work aims to advance the state-of-the-art in UAV control, paving the way for the development of more intelligent, robust, and autonomous UAV systems capable of operating in increasingly complex and dynamic environments [24,26,28].

### 4.2. Limitations of Traditional Linear Control Techniques

Traditional linear control techniques, while effective in many standard control applications, face inherent limitations when applied to the complex dynamics of UAVs [54]. These limitations stem from the fact that UAVs often exhibit nonlinear behaviors and interactions that cannot be adequately addressed by linear control methods. Some of the key limitations include:


**Inadequacy in Handling Nonlinear Dynamics**


Linear Approximation: Linear control techniques are based on linear approximations of system dynamics. However, UAVs often exhibit strongly nonlinear behaviors due to aerodynamic forces, changes in air density at different altitudes, and the nonlinear response of their propulsion systems [55].Limited Operational Range: Linear controllers are typically designed around a specific operating point, such as hovering. They tend to be less effective when the UAV operates outside this narrow range, such as during rapid maneuvers or in response to strong external disturbances, and may even fail [56].


**Oversimplification of Real-World Scenarios**


Ignoring External Factors: Linear controllers often do not account for varying external conditions such as wind gusts, temperature changes, or variable payloads, which can significantly affect UAV performance [57].Simplified Models: These controllers rely on simplified models that may not capture the full complexity of UAV dynamics, leading to suboptimal or unstable flight in real-world scenarios [58].


**Inability to Manage Coupling Effects**


Independent Control Assumption: Linear control techniques often assume that each control input affects only one output. However, in UAVs, especially multi-rotor systems such as quadrotors, there is significant coupling between controls, e.g., changing rotor speed to control roll might inadvertently affect pitch and yaw [59].Challenges in MIMO Systems: UAVs are often MIMO (Multiple Input, Multiple Output) systems where the interaction between multiple inputs and outputs needs to be managed simultaneously, a task for which traditional linear control methods are not well-suited [60].


**Lack of Adaptability and Robustness**


Fixed Parameters: Traditional linear controllers have fixed parameters and do not adapt to changes in the UAV’s dynamics or the environment. This lack of adaptability can lead to poor performance in changing conditions [61].Robustness Issues: They may not be robust against uncertainties or unmodeled dynamics, which are common in UAV operations, especially in complex or unstructured environments [62].


**Increased Complexity in Design and Tuning**


Complex Design for Multivariable Systems: Designing linear controllers for systems with multiple interdependent variables, such as UAVs, can be complex and time-consuming [29].Manual Tuning Limitations: These controllers often require manual tuning, which can be a laborious process and might not yield the best possible performance [63].

While traditional linear control techniques have been foundational in control theory and have applications in many areas, their limitations become apparent in the context of UAVs. As explained in Figure 10, the complexity, nonlinearity, and dynamic nature of UAVs call for more advanced control strategies that can adapt to changing conditions, handle nonlinear dynamics, and manage the coupling effects inherent in these systems.

### 4.3. Impact of Sensor Data Complexity on UAV Control

The integration and utilization of sensor data plays a critical role in the control and operation of UAVs [64]. However, the complexity of sensor data can significantly impact UAV control systems in various ways:


**Data Volume and Processing Challenges**


High Data Volume: UAVs often utilize a multitude of sensors such as cameras, LiDAR, GPS, gyroscopes, and accelerometers, which generate a vast amount of data. Processing this high volume of data in real time can be challenging, requiring robust and efficient data processing algorithms [29].Computational Load: The need to swiftly process and analyze complex sensor data places a considerable computational load on the UAV’s onboard systems, which could impact its operational efficiency and responsiveness [65].


**Sensor Fusion and Integration**


Complexity of Sensor Fusion: Integrating data from multiple sensors to form a cohesive understanding of the UAV’s environment (sensor fusion) is complex. It requires sophisticated algorithms to accurately combine and interpret disparate data sources, which can be challenging, especially in dynamically changing environments [66].Inconsistencies and Conflicting Data: Different sensors may provide conflicting information due to varied accuracy, resolution, or response times. Resolving these inconsistencies is crucial for accurate flight control and decision-making [67].


**Accuracy and Reliability Concerns**


Sensor Accuracy: The precision and accuracy of sensors directly affect the control and stability of UAVs. Inaccuracies in sensor readings can lead to incorrect control commands and potentially unstable flight [68].Reliability under Diverse Conditions: Sensors must be reliable under a wide range of operational conditions. For example, visual sensors might be less effective in low-light or foggy conditions, impacting the UAV’s ability to navigate and avoid obstacles [69].


**Real-Time Decision Making**


Delayed Response: The time taken to process complex sensor data can lead to delays in decision-making, which is critical for UAVs that need to respond quickly to environmental changes or obstacles [70].Autonomous Operations: For autonomous UAVs, the ability to make real-time decisions based on sensor data is crucial. Complex data can complicate the algorithms needed for autonomous navigation and task execution [71].


**Energy Consumption**


Delayed Responses: The time taken to process complex sensor data can lead to delays in decision-making, which is critical for UAVs that need to respond quickly to environmental changes or obstacles [72].Autonomous Operations: For autonomous UAVs, the ability to make real-time decisions based on sensor data is crucial. Complex data can complicate the algorithms needed for autonomous navigation and task execution [73].Increased Power Demand: Processing complex sensor data requires significant computational resources, which in turn increases energy consumption. This can reduce the UAV’s operational endurance and limit its range or mission duration [74].


**Calibration and Maintenance**


Calibration Complexity: Accurate sensor data depends on proper calibration. Complex sensor systems may require frequent and sophisticated calibration procedures, adding to the operational overhead [75].Maintenance Requirements: More complex sensor systems might have higher maintenance needs, impacting the UAV’s readiness and operational costs [76].

While sensors are indispensable for modern UAV operations, their complexity introduces several challenges. As shown in Figure 11, effectively managing these challenges involves developing advanced computational algorithms for real-time processing, improving sensor technologies for better accuracy and reliability, and optimizing UAV designs for efficient energy use. The ultimate goal is to ensure that UAVs can effectively interpret and respond to their environment, balancing the complexity of sensor data with the need for swift and accurate control.

## 5. Nonlinear Control Strategies: Sliding Surface-Based Control Strategies

Nonlinear control strategies are essential for highly nonlinear and coupled UAV systems. Linear control techniques are not suitable for UAVs because of nonlinearities and coupling effects. Nonlinear control strategies are designed to handle these nonlinearities and coupling effects, making them more effective for UAVs [76,77].

There are various nonlinear control strategies available for UAVs, including sliding mode control [76,77], fuzzy control [76], adaptive control [78], and neural network control [79,80]. Sliding mode control is one of the most popular nonlinear control techniques for UAVs. It provides robustness against parameter uncertainties and external disturbances. Fuzzy control is a nonlinear control technique that uses linguistic variables to control the system. It is robust against uncertainties and disturbances, and can handle the nonlinearities and coupling effects of UAVs. Adaptive control uses adaptive laws to adjust the control parameters based on the system’s dynamics, making it suitable for TRMS with varying operating conditions. Neural network control uses artificial neural networks to model the system and provide control.

### 5.1. First-Order Sliding Mode Control (SMC)

The foundation of SMC is VSS control theory, which operates under the tenet that in order to maintain the states of the system in sliding mode, the controller structure must continuously change in response to variations in the state variables. By using switching control with a high frequency, the SMC has a tendency to alter system dynamics [81,82,83]. The VSS control theory is the foundation for the idea of sliding mode control, which operates on the idea that the controller design should change continually in response to changes in the state variables in order to maintain the system states in sliding mode. There are two distinct parts to the controller design [34,84,85,86]. The sliding surface is assessed in the first portion using the system’s order as a basis.

The intended sliding surface is expressed as follows:(21)$$s\left(t\right)={\left(\frac{d}{d_{t}}+\lambda \right)}^{n}\;\int_{0}^{t}e_{\xi }\left(t\right)\;d_{t}.$$

Here, $e_{\xi }\left(t\right)$ is the observed monitored output with the new state as the state variable and s(t) is the chosen manifold used as the sliding surface. It is vital to consider the control rule that forces the control factors to their reference value while choosing the sliding surface [87,88]. The mathematical formulation of the control law is
(22)$$u=u_{eq}+u_{dis},$$
while
(23)$$u_{dis}=-ksign\left(s\right).$$

Here, the constant $k_{1}$, the sign function sign(s), and the equivalent controller $u_{eq}$ provide a fragmented control input as a result of a limited switching over the sliding surface. The movement of numerous control structures causes the controlled system’s trajectory to slide along a variety of planes while adhering to the switching condition. It has been noted that switching functions s(x), where *x* is either a scalar or vector, can be used during SMC to define the system structure [89,90,91]. A line on the phase plane represents the switching surface, denoted by s(x)=0.

The instability of a nonlinear system can be examined using the Lyapunov function and the theory of ODEs [92,93]. The ODE class theory calculates the system’s Lyapunov function, which has to be negative definite, and uses the results to confirm the stability of the system. The nonlinear system’s asymptotic stability is guaranteed by this necessary condition. However, an appropriate methodology for building functions for ODEs is lacking [94,95]. In practice, the sliding motion is observed to exist in the vicinity of the sliding surface, much like the frequency-switching phenomenon in Figure 12. The system’s nonlinear behavior will attempt to stray off the sliding surface, while the controller will push it to stay on course until it converges at the boundary layer or origin [96,97]. The power supply needs to be highly optimized because of the quick and abrupt changes in voltage patterns. Optimization methods can be used to achieve the required UAV output with steady settling and a predictable voltage pattern.

Chattering is a well-known phenomenon in SMC that can cause high-frequency oscillations in the control signal. These oscillations can lead to actuator failures due to excessive wear and tear, and can result in degraded control performance [98,99,100].

In one study [51], SMC was used to control a UAV for target tracking and hovering tasks. The results showed that SMC was able to track the target accurately and robustly even in the presence of disturbances and model uncertainties. SMC is a popular nonlinear control strategy for UAVs due to its robustness against disturbances and uncertainties. In SMC, a sliding surface is defined such that the dynamics of the system are constrained to remain on the surface. Several studies have used sliding mode control for UAVs. In one study [101], SMC was used for the trajectory tracking control of a UAV. The proposed controller was able to achieve accurate and robust tracking even in the presence of disturbances and modeling uncertainties. In another study [102], SMC was used for the attitude control of a UAV. The proposed controller was able to achieve robust performance in the presence of external disturbances and parameter uncertainties. UAV systems are highly nonlinear and coupled, making them challenging to control using traditional linear control methods. SMC has limitations around obtaining the required response, making adaptation law-based SMC variants better for complex MIMO models.

#### Remarks

SMC is known for being insensitive to external disturbances and nonlinearities. Observer-based SMC is a good option for reducing chattering while maintaining the invariance property of SMC; however, observer-based SMC is not suitable for dealing with parametric uncertainties or for maintaining tracking accuracy. Adaptive sliding mode control (ASMC) is more robust than either conventional SMC or observer-based SMC. Therefore, ASMC can be used instead of SMC to achieve adaption and remove chattering.

### 5.2. Backstepping (Recursive) Structure-Based Control Strategies

Backstepping (recursive) structure-based control strategies are an important class of nonlinear control techniques that have gained significant attention in recent years. The importance of these strategies lies in their ability to handle complex nonlinear control problems with high precision and efficiency. Backstepping control strategies are important for a number of specific reasons:Nonlinear control: Backstepping control strategies are designed specifically for nonlinear control problems, which are common in many engineering applications. These strategies provide a framework for designing controllers that can handle nonlinear dynamics and uncertainties while ensuring stability and convergence.Recursive structure: Backstepping control strategies have a recursive structure that allows for the systematic design of control laws. This structure provides a natural way to build up the control law step-by-step, starting from the highest-order states and working downwards. This recursive approach simplifies the control design process, making it easier to develop complex controllers.Lyapunov stability analysis: Backstepping control strategies are typically designed using Lyapunov stability analysis, which provides a rigorous mathematical framework for assessing the stability and convergence properties of the control system. This analysis ensures that the designed controller is stable and that the system state converges to the desired state in a finite time.Robustness: Backstepping control strategies are inherently robust against disturbances and uncertainties, as they are designed to handle nonlinear dynamics and uncertainties in a systematic way. This robustness makes them particularly useful in applications where the system model is uncertain or poorly known.Performance: Backstepping control strategies can achieve high control performance because they are designed to optimize a performance criterion based on the system dynamics and control objectives. This optimization ensures that the control law is designed to achieve the desired control performance while ensuring stability and robustness.

Backstepping control is a type of nonlinear control technique that has gained attention for its effectiveness in controlling UAVs. The backstepping control approach is particularly suitable for UAVs because it can handle complex the nonlinear dynamics and uncertainties that are present in these vehicles. Backstepping control has been used in various UAV applications, including trajectory tracking, altitude control, and stabilization. One of the most significant advantages of backstepping control is that it can provide robust performance in the presence of disturbances and uncertainties. Additionally, it allows for the design of a control law that ensures the stability of the system while achieving the desired tracking performance.

Several studies have reported the effectiveness of backstepping control for UAV applications. For example, in one study [103], backstepping control was used for the trajectory tracking of a UAV, with the results showing improved tracking accuracy compared to other control methods. In another study [104,105], backstepping control was used for the altitude control of a fixed-wing UAV, and the results showed good performance even in the presence of wind disturbances.

#### 5.2.1. Mathematical Formulation of Backstepping Control

Backstepping is a recursive control design technique that is particularly well-suited for nonlinear systems such as UAVs. For TRMS UAVs, we can consider the control of the pitch (θ) and yaw (ψ) angles using backstepping.

The dynamic equations for the pitch and yaw angles of a TRMS UAV can be represented as follows:(24)$$\begin{array}{cc} I_{xx}\ddot{\theta } & =\tau_{\theta }-\left(I_{zz}-I_{yy}\right)\dot{\psi }\dot{\phi } \end{array}$$(25)$$\begin{array}{cc} I_{yy}\ddot{\psi } & =\tau_{\psi }-\left(I_{xx}-I_{zz}\right)\dot{\phi }\dot{\theta } \end{array}$$
where $I_{xx}$, $I_{yy}$, and $I_{zz}$ are the moments of inertia about the principal axes, $\tau_{\theta }$ and $\tau_{\psi }$ are the respective control torques for the pitch and yaw, and $\dot{\phi }$ is the roll rate.

To design a backstepping controller, we introduce virtual control inputs and Lyapunov functions to recursively stabilize the system. The control law for the pitch and yaw angles can be formulated as follows:(26)$$\begin{array}{cc} \tau_{\theta } & =-k_{\theta }\left(\theta -\theta_{\mathrm{ref}}\right)-k_{\dot{\theta }}\left(\dot{\theta }-{\dot{\theta }}_{\mathrm{ref}}\right) \end{array}$$(27)$$\begin{array}{cc} \tau_{\psi } & =-k_{\psi }\left(\psi -\psi_{\mathrm{ref}}\right)-k_{\dot{\psi }}\left(\dot{\psi }-{\dot{\psi }}_{\mathrm{ref}}\right) \end{array}$$
where $k_{\theta }$, $k_{\dot{\theta }}$, $k_{\psi }$, and $k_{\dot{\psi }}$ are the control gains and $\theta_{\mathrm{ref}}$, ${\dot{\theta }}_{\mathrm{ref}}$, $\psi_{\mathrm{ref}}$, and ${\dot{\psi }}_{\mathrm{ref}}$ are the desired pitch and yaw angles and their corresponding rates.

The Lyapunov functions are chosen to ensure the stability of the closed-loop system; the recursive nature of backstepping allows for the design of controllers for higher-order systems by sequentially stabilizing subsystems. By applying this backstepping control law, the pitch and yaw angles of the TRMS UAV can be effectively controlled, ensuring stability and desired performance.

#### 5.2.2. Remarks

Backstepping control has been successfully applied to various physical systems, including UAVs, robots, and mechanical systems. However, as with any control strategy, backstepping control has some limitations. Backstepping control can be complex to design and implement, especially for systems with many states or highly nonlinear dynamics. The controller design involves multiple steps, and the resulting controller may be difficult to analyze and understand. Backstepping control assumes that the system model is known, which may not be true in practice. If there are uncertainties or disturbances in the system, the controller may not perform well and may even become unstable. Backstepping control relies on accurate measurements of the system states. If there is measurement noise, the controller’s performance may be degraded. The performance of the backstepping controller depends on the choice of tuning parameters, such as gains and feedback coefficients. Tuning these parameters can be time-consuming and may require expert knowledge. Backstepping control may not be robust against changes in the system parameters or external disturbances. This can be a problem in practical applications where the system may experience uncertain or varying environmental conditions. Finally, backstepping control is not a panacea and may not be suitable for all types of MIMO channel systems or applications. Proper consideration of these limitations and appropriate modifications may be necessary for successful application in specific situations.

### 5.3. Feedback Linearization Control

Instead of linearizing via small angle approximation, this control paradigm transforms the UAV’s nonlinear dynamics into linear corresponding dynamics via feedback. This control method has been used by several researchers to create UAV flight controllers [53,106,107,108,109,110,111,112,113,114]. Due to the nonlinear nature of UAV dynamics, this method is well suited for removing nonlinearities, after which linear control theory can be employed to develop a flight controller. Feedback linearization eliminates the system’s nonlinearity by transforming the variables and choosing the right input. Then, using an outer feedback loop, a new control input *v* is computed using linear control theory. The transformation must be diffeomorphic in order to guarantee that it is comparable to the original system. Zero dynamics, or states that cannot be observed from the system output, are present in transformed systems, and may cause instability; therefore, the stability of these zero dynamics should be maintained. Due to the high-order Lie derivatives used in the linearization process, this technique is quite susceptible to outside disturbances. The fact that feedback linearization only applies to a particular class of nonlinear systems that satisfy a particular involutivity criterion is another disadvantage of this technique [115,116,117]. Zero dynamics, also known as internal dynamics, are crucial in determining the stability of the control system. Technically speaking, zero dynamics are the internal system states that are not visible from the system output. These dynamics could be unstable and harmful for the internal states, as they can become unbounded. It is necessary to take steps to guarantee that these internal dynamics do not lead to stability issues, and these unobservable states should at the very least be stable or controllable; see [53,115,118] for further information on zero dynamics.

A comprehensive study of accurate linearization for a UAV model has been presented in [53,119]. This study showed that static feedback cannot solve the decoupling issue of the UAV paradigm. Instead, it was discovered that dynamic feedback can transform a system into one that is linear, disconnected, and non-interactive. Furthermore, dynamic feedback can guarantee the controllability of the altered system. On the basis of attitude representation, an exact input-output linearization of the UAV model was developed in [107,120]. The linearization of the *z* and (x,y) states was performed separately, resulting in what is known as quasi-static feedback linearization [120,121]. A similar investigation of accurate linearization can be found in [53,106,122], while the inclusive mathematical design and feedback linearization approach can be found in [109]. By adding successive Lie derivative terms, UAV nonlinear dynamics were converted to linear dynamics. Transformed linear systems have derivative terms up to the third order, which renders the control strategy extremely susceptible to external errors and sensor noise. Similar work with feedback linearized controllers has been reported in [11,53]. In [106,108], along with a fix for rotor failure, the authors proposed a feedback linearization-based aircraft controller for the problem of trajectory tracking. The same method was used to achieve steady rotational motion across the *z*-axis in the event of rotor failure. A similar investigation was presented in [113,115], where several fault scenarios involving UAV flights were looked into. Attitude stabilization was accomplished by combining feedback linearization control and an inner loop.

#### 5.3.1. Mathematical Formulation

Feedback linearization is a control design technique that aims to transform a nonlinear system into a linear one through a suitable change of coordinates. For a TRMS UAV, the pitch (θ) and yaw (ψ) angles can be controlled using feedback linearization.

The dynamic equations for the pitch and yaw angles of a TRMS UAV can be represented as shown below.
(28)$$\begin{array}{cc} I_{xx}\ddot{\theta } & =\tau_{\theta }-\left(I_{zz}-I_{yy}\right)\dot{\psi }\dot{\phi } \end{array}$$
(29)$$\begin{array}{cc} I_{yy}\ddot{\psi } & =\tau_{\psi }-\left(I_{xx}-I_{zz}\right)\dot{\phi }\dot{\theta } \end{array}$$

To apply feedback linearization, we introduce a change of coordinates to linearize the system. We first define the following virtual control inputs.
(30)$$\begin{array}{cc} v_{1} & =\dot{\theta }-{\dot{\theta }}_{\mathrm{ref}} \end{array}$$
(31)$$\begin{array}{cc} v_{2} & =\dot{\psi }-{\dot{\psi }}_{\mathrm{ref}} \end{array}$$

The transformed system can be written as follows:(32)$$\begin{array}{cc} {\dot{v}}_{1} & =-k_{\theta }\left(v_{1}-{\dot{\theta }}_{\mathrm{ref}}\right) \end{array}$$(33)$$\begin{array}{cc} {\dot{v}}_{2} & =-k_{\psi }\left(v_{2}-{\dot{\psi }}_{\mathrm{ref}}\right) \end{array}$$
where $k_{\theta }$ and $k_{\psi }$ are the control gains and ${\dot{\theta }}_{\mathrm{ref}}$ and ${\dot{\psi }}_{\mathrm{ref}}$ are the desired pitch and yaw rates.

The control torques $\tau_{\theta }$ and $\tau_{\psi }$ can then be determined using the inverse dynamics.
(34)$$\begin{array}{cc} \tau_{\theta } & =I_{xx}\ddot{\theta }+\left(I_{zz}-I_{yy}\right)\dot{\psi }\dot{\phi } \end{array}$$
(35)$$\begin{array}{cc} \tau_{\psi } & =I_{yy}\ddot{\psi }+\left(I_{xx}-I_{zz}\right)\dot{\phi }\dot{\theta } \end{array}$$

Substituting the expressions for $\ddot{\theta }$ and $\ddot{\psi }$ from the transformed system into the inverse dynamics equations yields the control inputs required to achieve the desired pitch and yaw rates.

By applying feedback linearization, the pitch and yaw angles of the TRMS UAV can be effectively controlled by linearizing the system and designing control inputs to track the desired rates. This approach offers a systematic way of handling the nonlinearities and coupling effects present in the UAV dynamics, ensuring stable and precise control of the UAV’s motion.

#### 5.3.2. Remarks

Feedback linearization is a useful technique for changing a nonlinear system into a matching linear system; however, as already discussed, it suffers from instability zero dynamics. Linearization of a quadrotor produces zero dynamics, as its nonlinear dynamics are underactuated. By placing two consecutive integrators in the direction of each control input, as in [123,124], the issue of unstable zero dynamics can be handled. The complexity of zero dynamics requires in-depth analysis. The need for full-state statistics is one of the biggest drawbacks of the feedback linearization approach, necessitating the design of a separate observer/estimator for estimation of the system’s states. In [123,125], observer design and feedback linearization for UAVs were both achieved. The third derivative of the output states is required for feedback linearization of a UAV model; however, in this study the observer model was employed to constrain the third derivative, making the control technique more acceptable for use with nonlinear systems.

The UAV model’s parametric uncertainties may lead to performance problems. This calls for the application of adaption laws and control strategies that can lessen parametric uncertainty and accommodate changes in parametric values. A linearized adaptive feedback controller was created in [126] and a thorough structure was provided. A similar investigation of an adaptive feedback linearized controller architecture was presented in [123,127]. The adaptive technique decreased the tracking error and enhanced the controller’s performance. Through adaptation, the control parameters can be changed, improving the overall performance of the control system.

#### 5.3.3. Remarks

When the system variables are known and there is no related uncertainty, feedback linearization performs well; on the other hand, UAVs may encounter significant problems due to their own parametric instabilities and feedback linearization-based controllers. In order to completely solve the aforementioned issue, an adaptive intelligent technique must be used along with traditional feedback linearization. Control performance can be greatly enhanced by online updating and by employing nonlinear approximators to approximate the unknown parameters.

### 5.4. Model Predictive Control (MPC)

MPC is a nonlinear control technique that can optimize the control inputs over a finite time horizon. MPC has been used for UAVs in control applications such as surveillance and mapping tasks. In one study [128,129,130,131,132], MPC was used to control a fixed-wing UAV for surveillance tasks. The results showed that MPC was able to improve the accuracy and efficiency of the UAV during the surveillance task [129,133]. MPC has a prediction advantage over LQR control because the latter cannot foresee future control inputs [134,135]. Second, although LQR only offers optimal control for linearized plant model systems, MPC may be used for nonlinear systems as well. Without a precise prediction model, the MPC controller’s algorithms cannot achieve extraordinary stability and performance [135]. As a result, if the prediction model is imprecise, the system may become unstable, and creating such a model demands an expensive control design effort. In addition, the capacity of adaptive MPC to estimate uncertainty makes it more reliable and useful for optimizing the control of a UAV. Future research could concentrate on developing an adaptive MPC aircraft controller, as this area of research into UAV control has seen little study.

#### 5.4.1. Mathematical Formulation of MPC

Model Predictive Control (MPC) is an advanced control strategy that uses a predictive model of the system to optimize control inputs over a finite horizon. For a TRMS UAV, the pitch (θ) and yaw (ψ) angles can be controlled using MPC.

The dynamic equations for the pitch and yaw angles of a TRMS UAV are represented in following equations.
(36)$$\begin{array}{cc} I_{xx}\ddot{\theta } & =\tau_{\theta }-\left(I_{zz}-I_{yy}\right)\dot{\psi }\dot{\phi } \end{array}$$
(37)$$\begin{array}{cc} I_{yy}\ddot{\psi } & =\tau_{\psi }-\left(I_{xx}-I_{zz}\right)\dot{\phi }\dot{\theta } \end{array}$$

In MPC, we formulate a predictive model of the system and optimize a control sequence to minimize a cost function over a finite prediction horizon. We can define the state vector as shown below.
(38)$$\begin{array}{cc} x & =\left[\begin{array}{c} \theta \\ \dot{\theta }\\ \psi \\ \dot{\psi } \end{array}\right] \end{array}$$

The state space representation of the system can be written as follows:(39)$$\begin{array}{cc} \dot{x} & =Ax+Bu \end{array}$$(40)$$\begin{array}{cc} y & =Cx \end{array}$$
where
(41)$$\begin{array}{cc} A & =\left[\begin{array}{cccc} 0 & 1 & 0 & 0\\ 0 & 0 & 0 & 0\\ 0 & 0 & 0 & 1\\ 0 & 0 & 0 & 0 \end{array}\right],\;B=\left[\begin{array}{cc} 0 & 0\\ \frac{1}{I_{xx}} & 0\\ 0 & 0\\ 0 & \frac{1}{I_{yy}} \end{array}\right],\;C=\left[\begin{array}{cccc} 1 & 0 & 0 & 0\\ 0 & 0 & 1 & 0 \end{array}\right]. \end{array}$$

The control inputs *u* can be represented as follows:(42)$$\begin{array}{cc} u & =\left[\begin{array}{c} \tau_{\theta }\\ \tau_{\psi } \end{array}\right]. \end{array}$$

In MPC, the control inputs are optimized by solving the following optimization problem at each time step:(43)$$\begin{array}{cc} \min_{u}\; & J(x,u) \end{array}$$(44)$$\begin{array}{cc} s.t.\; & x(k+1)=Ax(k)+Bu(k) \end{array}$$(45)$$\begin{array}{cc} \; & u_{\min}\leq u\leq u_{\max} \end{array}$$
where J(x,u) is the cost function and $u_{\min}$ and $u_{\max}$ are the minimum and maximum torque limits, respectively.

By solving this optimization problem at each time step, MPC computes the optimal control inputs that minimize the cost function while satisfying the system dynamics and control constraints. This enables precise and robust control of the pitch and yaw angles of TRMS UAVs even in the presence of uncertainties and disturbances.

#### 5.4.2. Remarks

MPC has several benefits for controlling highly coupled wind rotor MIMO systems such as UAVs. First, MPC is capable of accurately tracking a desired trajectory for the UAV. This is important for UAVs that need to follow specific flight paths or perform complex maneuvers. Second, MPC is robust against uncertainties and disturbances in the UAV’s dynamics, making it a suitable control method for UAVs operating in dynamic and uncertain environments. Third, MPC can easily handle constraints on the UAV’s inputs and states. This is important for UAVs that need to operate within certain physical limitations, such as maximum velocity, altitude, or acceleration. Third, MPC can simultaneously optimize multiple objectives, such as minimizing energy consumption while maximizing tracking performance. This makes MPC a versatile control method that can adapt to different operating conditions and environmental factors. Fourth, MPC can be easily integrated with other systems, such as sensors, communication networks, and mission planning software, making it a suitable control method for UAVs in complex and dynamic environments. Fifth, MPC can be easily adapted to different UAV configurations, allowing it to handle different operating conditions and environmental factors. Finally, MPC produces control signals that are smooth and continuous, which reduces chattering in the control signal. This results in smoother control actions and reduces wear and tear on the UAV’s mechanical components.

While MPC has many benefits for controlling highly coupled wind rotor MIMO systems such as UAVs, there are also limitations to this control method. First, MPC requires solving optimization problems at each time step, which can be computationally intensive and may require high-performance computing resources. This can be a limitation for UAVs that require fast and real-time control responses. Second, MPC requires tuning of several parameters, such as the prediction horizon, control horizon, and weighting factors for the optimization objectives. If these parameters are not tuned correctly, the control performance can be degraded. This can be a limitation for UAVs that operate in changing environments where the optimal parameter values may vary. Third, MPC is dependent on the accuracy of the mathematical model used to describe the system dynamics. If the model is not accurate, the control performance can be compromised. This can be a limitation for UAVs that have complex and highly nonlinear dynamics. Fourth, MPC relies on predicting the future state of the system, which can lead to a delayed response to disturbances or changes in the environment. This can be a limitation for UAVs that require fast and immediate responses to changes in the environment. Fifth, MPC is a complex control method that requires a significant amount of technical expertise to implement and maintain. This can be a limitation for UAVs that have limited resources or operate in remote locations. Finally, the control decisions made by MPC can be difficult to interpret or explain, making it challenging to diagnose or troubleshoot control system issues.

While MPC has many benefits for controlling highly coupled MIMO systems such as UAVs, the above limitations must be carefully considered when selecting a control method for a particular application. Proper implementation and tuning are critical to ensuring the control system’s stability, tracking performance, and robustness in the presence of disturbances and uncertainties.

### 5.5. Contribution of Nonlinear Control Strategies for UAVs

Nonlinear control strategies offer a diverse range of approaches that can be effectively applied to UAV systems, each with its own unique advantages and capabilities. Among the nonlinear control strategies discussed in this review, MPC stands out for its ability to handle complex dynamical systems and optimize control inputs over a predictive horizon, making it particularly suitable for UAVs operating in dynamic environments. SMC, on the other hand, excels in robustness against model uncertainties and external disturbances, ensuring reliable performance in challenging conditions. Adaptive and neural network-based control strategies offer flexibility and adaptability by learning from the UAV’s environment, enabling autonomous decision-making and adaptation to varying operating conditions. Finally, feedback linearization provides a systematic approach to transforming nonlinear dynamics into linear ones, facilitating the design of linear controllers that can effectively stabilize and control UAV systems. Each of these nonlinear control strategies has been tailored to address specific challenges encountered in UAV applications, offering a versatile toolkit for enhancing UAV performance, autonomy, and reliability.

## 6. Hybrid Control Strategies

Hybrid control theory is a branch of modern control theory that focuses on systems demonstrating both continuous and discrete behaviors. These systems, referred to as hybrid systems, present a unique challenge due to their combination of continuous dynamics (such as those seen in traditional control systems with smooth and continuous changes) and discrete events (such as switching between different operational modes or the occurrence of events triggering abrupt changes in behavior) [136,137].

In hybrid systems, the interaction between continuous dynamics and discrete events is fundamental. Continuous dynamics represent the gradual evolution of system states over time, while discrete events introduce sudden changes or switches in the system’s behavior or mode of operation. For example, in a hybrid control system for an electric vehicle, continuous dynamics might govern the motor’s speed and acceleration, while discrete events could include switching between driving modes (e.g., electric-only mode and hybrid mode) or activating regenerative braking when certain conditions are met [138].

Hybrid control theory aims to develop modeling, analysis, and control techniques tailored to these complex systems considering both continuous and discrete aspects. It addresses challenges such as the design of controllers that can handle both continuous dynamics and discrete events, thereby ensuring stability and performance in the face of mode switches or sudden disturbances and optimizing system behavior under various operating conditions. Overall, hybrid control theory provides a framework for understanding and effectively managing the intricate behavior of hybrid systems in various engineering applications [136]. Developments in hybrid control are provided in Table 3.

UAVs represent a sophisticated control system utilized within the UAV’s operations. In this context, hybrid control plays a crucial role due to its capability to offer a robust and adaptable control strategy. In a hybrid control setup, various control methodologies are integrated to enhance performance and ensure stability. This approach proves particularly beneficial in UAV applications in light of the diverse and dynamic environments in which these systems operate. By amalgamating different control techniques, the UAV system can effectively adjust to varying conditions and uphold stability, thereby enhancing its overall operational efficiency and reliability [141,142].

### 6.1. Adaptive Sliding Mode Control (ASMC)

Adaptive sliding mode control (ASMC) is a variant of SMC that incorporates adaptive techniques to improve the control performance of nonlinear MIMO (multiple-input multiple-output) systems. For several reasons, ASMC is generally considered to be better than SMC for nonlinear MIMO systems. Nonlinear MIMO systems often have complex and uncertain dynamics that can degrade the performance of conventional control strategies such as SMC. ASMC incorporates adaptive mechanisms that allow the control parameters to be adjusted in real time based on the system’s current operating conditions, making it more robust against model uncertainties [143]. ASMC can achieve better tracking accuracy compared to SMC for nonlinear MIMO systems. This is because ASMC uses a model reference adaptive control (MRAC) approach that provides more accurate tracking of the desired trajectory while maintaining robustness against system uncertainties [144]. SMC can suffer from chattering, which is a phenomenon characterized by high-frequency oscillations in the control signal. ASMC can reduce chattering by incorporating adaptive gain scheduling, which adjusts the control gain in real time to reduce the effects of chattering and improve control performance [145]. ASMC can achieve the same control performance as SMC with lower control effort, which can be beneficial for systems with limited actuator capacity or those that operate in harsh environments [143,145]. Overall, ASMC offers several advantages over SMC for nonlinear MIMO systems, including improved robustness, better tracking accuracy, and reduced chattering. However, the design and implementation of ASMC can be more complex than SMC, and it may require higher computational overhead. Adaptive sliding mode control (ASMC) is a nonlinear control strategy that has been applied to UAVs for effective control in the presence of uncertainties and disturbances. ASMC has been shown to be effective in controlling UAV in various scenarios. In one study [146], an ASMC approach was used for the position and attitude control of a UAV. The results showed improved control performance compared to traditional linear control methods. In another study [147], an ASMC approach was used for the stabilization of a UAV in the presence of parameter uncertainties, with the results showing good robustness and disturbance rejection. The main advantage of ASMC is its ability to handle uncertainties and disturbances by using a sliding mode control law that drives the system to a sliding surface, where it can maintain stability and robustness. Additionally, the use of an adaptive law allows the system to adapt to changing conditions and uncertainties, improving the overall control performance. Although ASMC is an efficient control strategy, the convergence time must be faster and the system should move toward stability more quickly.

#### Remarks

While adaptive SMC is more robust than conventional SMC, it has slower convergence; therefore, ATSMC can be used to achieve faster convergence along with chattering removal. According to the above discussion, using ATSMC with an adaptation design can result in the best control performance. Furthermore, ATSMC offers a simple nonlinear design procedure for flight controllers, which leads to lower computational costs compared to other methods. Overall, the findings highlight the benefits of SMC and the use of ATSMC for flight control to achieve better control performance, faster convergence, and reduced computational cost, particularly with adaptive designs.

### 6.2. Adaptive Fast-Terminal Sliding Mode Control (AFTSMC)

UAVs are a popular benchmark system in the field of control engineering. Adaptive fast-terminal sliding mode control (AFTSMC) is a robust control technique that has been applied to UAVs to achieve improved control performance. The design procedure of AFTSMC for UAVs involves the following steps:System modeling: The first step is to develop a mathematical model of the UAV. The model should include the dynamics of both the rotor and the platform, and should be expressed in state space form.Control objective: The control objective is to design a control law that can stabilize the UAV at a desired position and orientation while rejecting disturbances and uncertainties.Sliding mode control: AFTSMC is based on sliding mode control (SMC), which involves the design of a sliding surface that ensures fast convergence to the desired state. The sliding surface should be designed such that its derivative is negative definite and the system trajectory approaches it asymptotically.Terminal sliding mode control: In addition to SMC, AFTSMC incorporates terminal sliding mode control (TSMC) to achieve faster convergence to the desired state. TSMC involves the design of a terminal sliding surface which is reached in a finite time and remains stable thereafter.Adaptive control: AFTSMC incorporates adaptive control to account for uncertainties in the system parameters. Adaptive control involves the design of an adaptation law that updates the control gains in real time based on the estimated system parameters.Design of control gains: The final step is to design the control gains such that the sliding surface and the terminal sliding surface are reached in a finite time and the system remains stable thereafter. The control gains can be tuned using simulation or experimental data to achieve optimal performance.

The design procedure of AFTSMC for UAVs involves the integration of SMC, TSMC, and adaptive control to achieve robust and efficient control performance. AFTSMC is an improvement over ASMC that addresses some of the limitations of the latter. Among the main advantages of AFTSMC over ASMC, it converges to the sliding surface much faster due to the addition of a terminal sliding mode surface that has a higher convergence rate. This means that AFTSMC can achieve better tracking performance in a shorter time. In addition, AFTSMC is more robust against parameter uncertainties and disturbances compared to ASMC, as the terminal sliding mode surface provides guarantees convergence within a finite time even in the presence of disturbances and parameter uncertainties [148]. Finally, AFTSMC reduces chattering compared to ASMC. Chattering is a phenomenon in which the control signal switches rapidly between different values; this can cause wear and tear on the actuators, and reduces the overall system performance [149]. Adaptive fast-terminal sliding mode control (AFTSMC) is a nonlinear control technique that has been applied to UAVs for effective control in the presence of uncertainties and disturbances. AFTSMC is an extension of the traditional sliding mode control approach which incorporates a terminal sliding mode to improve the speed and accuracy of the system’s response. Several studies have investigated the use of AFTSMC for UAV control. In [149], AFTSMC was used for the attitude control of a UAV in the presence of uncertainties and external disturbances. The results showed improved control performance compared to traditional sliding mode control approaches. In another study [148] where AFTSMC was used for the position control of a UAV, the results again showed good robustness and disturbance rejection.

The key advantage of AFTSMC is its ability to achieve fast and accurate tracking of the desired trajectory even in the presence of uncertainties and disturbances. The terminal sliding mode allows the system to converge to a desired state in a finite time, while the adaptive law ensures that the system can adapt to changing conditions and uncertainties.

#### 6.2.1. Remarks for AFTSMC, ASMC, and SMC

Sliding surface-based control strategies are a class of control methods that rely on the design of a sliding surface, which is a function of the system state that is used to enforce a desired behavior or trajectory of the system. Several sliding surface-based control strategies, such as sliding mode control, adaptive sliding mode control, and adaptive fast terminal sliding mode control, have been reviewed for use in controlling highly coupled rotor MIMO systems such as UAVs. General remarks that apply to all sliding surface-based control strategies are provided below:


**Limitations**


While sliding surface-based control has several benefits for twin-rotor MIMO systems such as UAVs, it has several limitations as well. These include:Control signal chattering: Although first-order SMC can reduce chattering compared to higher-order sliding mode control techniques, it can still generate high-frequency oscillations in the control signal. This can cause wear and tear on mechanical components, decrease the lifespan of the vehicle, and lead to suboptimal control performance.Parameter sensitivity: First-order SMC can be sensitive to changes in the system parameters, such as the mass and moment of inertia of the UAV. This can lead to poor control performance or even instability if the parameters are not accurately known or change during operation.Simplicity of implementation: First-order SMC is relatively easy to implement and does not require extensive tuning of control parameters. This can reduce the development time and cost of UAV control systems.Reduced tracking accuracy: First-order SMC may not provide the same level of tracking accuracy as other control techniques such as model predictive control or linear quadratic regulator. This can be a limitation in applications where precise tracking of a desired trajectory is critical.Limited applicability: First-order SMC may not be suitable for all types of UAVs or operating conditions; for example, it may not be effective for highly dynamic systems with rapid changes in speed or acceleration.Limited convergence rate: The adaptation process in ASMC can lead to slower convergence rates compared to traditional sliding mode control techniques. This can be a limitation for UAVs that need to respond quickly to changes in their environment.Sensitivity to modeling errors: ASMC can be sensitive to modeling errors, which can lead to poor control performance or even instability. This is because the adaptation process relies on accurate knowledge of the system dynamics.Limited applicability: ASMC may not be suitable for all types of UAVs or operating conditions. For example, it may not be effective for highly nonlinear systems or systems with significant time delays.Chattering: A common issue with sliding surface-based control strategies is chattering, which involves high-frequency oscillation of the control signal around the desired value. Chattering can cause mechanical wear and tear in the actuators as well as noise and vibration. Techniques such as the use of saturation functions or the introduction of a switching gain can reduce the effect of chattering.Non-smoothness: Sliding surface-based control strategies are non-smooth, which means that the control signal can switch abruptly between different values. This non-smoothness can cause difficulties in the implementation of the control law, and can introduce high-frequency noise and vibration. Careful consideration of the physical limitations of the actuators and sensors is required to ensure that the control law can be implemented smoothly.Model dependency: The performance of sliding surface-based control strategies is highly dependent on the accuracy of the system model. Errors in the model can lead to poor performance or instability. Techniques such as adaptive control or model predictive control can be used to address this issue.Potential for actuator wear: The high-frequency control actions generated by TSMC can potentially cause wear and tear on the UAV’s mechanical components, such as its actuators. This can lead to increased maintenance costs and reduced system reliability over time.

AFTSMC may require the use of specialized hardware or software, such as high-speed processors or real-time operating systems. This can add to the complexity and cost of implementing the control technique. Adaptive fast-terminal sliding mode control (AFTSMC) is a relatively new control technique that has shown promising results in various nonlinear control applications. However, as with any control technique, it has its limitations.

One limitation of AFTSMC is that it can suffer from chattering, which is a phenomenon in which the control signal oscillates rapidly around the desired value. This can result in high-frequency noise as well as potential wear and tear on the mechanical components of the system. Several researchers have proposed modifications to AFTSMC to reduce chattering, such as adding a boundary layer to the sliding mode control or using fuzzy logic to dynamically adjust the sliding mode gain [149,150].Another limitation of AFTSMC is that it can be sensitive to model uncertainties and disturbances. While AFTSMC is designed to be adaptive to such uncertainties, in practice there may be situations where the uncertainties are too large or the adaptation process is too slow to compensate adequately. Several researchers have proposed modifications to AFTSMC to improve its robustness, such as using disturbance observers or incorporating online learning algorithms [151,152].

Sliding surface-based control strategies offer a powerful tool for controlling highly coupled rotor MIMO systems such as UAVs. While they provide robustness to disturbances and uncertainties, careful consideration of chattering, non-smoothness, and model dependency is required to ensure the successful implementation of the control law.


**Benifits of AFTSMC over SMC, TSMC, and ASMC**


AFTSMC is a robust control technique that has several advantages over other sliding surface-based hybrid controllers for UAVs. Among the benefits of AFTSMC are the following:Fast convergence: AFTSMC combines SMC and TSMC to achieve fast convergence to the desired state. TSMC enables the system trajectory to reach the sliding surface in a finite time, while SMC ensures that the sliding surface is stable thereafter. This results in faster convergence compared to other sliding surface-based hybrid controllers.Robustness: AFTSMC incorporates adaptive control to account for uncertainties in the system parameters. The adaptation law updates the control gains in real time based on the estimated system parameters, which enhances the robustness of the controller. This makes AFTSMC more effective in dealing with uncertainties and disturbances compared to other hybrid controllers.Chattering reduction: Chattering is a common problem in sliding mode control, resulting in high-frequency oscillations in the control signal. AFTSMC reduces chattering by incorporating a fast terminal sliding surface, which reduces the time spent on the sliding surface, and consequently the amplitude of the oscillations.Reduced control effort: AFTSMC reduces the control effort required to stabilize the UAV. Both the sliding surface and the terminal sliding surface are designed to minimize the control effort required to maintain stability, which reduces wear and tear on the system.

Overall, AFTSMC combines the benefits of SMC, TSMC, and adaptive control to achieve faster convergence, enhanced robustness, reduced chattering, and reduced control effort compared to other sliding surface-based hybrid controllers for UAVs.

#### 6.2.2. Remarks

Hybrid control is a control technique that combines multiple control strategies to achieve better control performance. Compared to other sliding surface-based controllers for UAVs, hybrid control has several advantages. Hybrid control has a number of benefits over AFTSMC and other sliding surface controllers. Hybrid control for UAVs can improve the robustness of the controller by combining multiple control strategies that can handle different types of uncertainties and disturbances. For example, hybrid control can combine adaptive control with model predictive control to achieve robust control performance in the presence of model uncertainties and disturbances. In addition, hybrid control can improve the tracking performance of the controller by combining different control strategies that are optimized for different aspects of the control problem. For example, hybrid control can combine feed-forward control with feedback control to achieve better tracking performance while minimizing control effort. Hybrid control can reduce chattering by combining different control strategies that can mitigate chattering in different ways. For example, hybrid control can combine sliding mode control with integral action to reduce chattering while maintaining fast convergence. Finally, hybrid control is a flexible control technique that can be customized to suit specific control problems. By combining different control strategies, hybrid control can be tailored to optimize control performance while taking into account system constraints and other requirements.

### 6.3. Adaptive Backstepping Control

Adaptive backstepping control is a nonlinear control approach that is used to design control systems for complex dynamic systems such as UAVs. It has the ability to handle parametric uncertainties and external disturbances effectively. By using adaptive backstepping control, UAVs can achieve better tracking performance, stability, and robustness. Several studies have proposed and investigated the use of adaptive backstepping control for UAVs. For instance, in [153] an adaptive backstepping control algorithm was proposed for UAVs with parametric uncertainties and external disturbances. The proposed algorithm was effective in tracking control and disturbance rejection. Another study [154] proposed an adaptive backstepping control method for UAVs with input saturation and external disturbances. The proposed control method was shown to achieve better tracking performance compared to traditional backstepping control. Moreover, in [155] an adaptive backstepping control method was proposed for UAVs with actuator saturation and external disturbances. The proposed method was shown to achieve improved tracking performance and robustness. Adaptive backstepping control has several benefits for controlling highly coupled twin-rotor MIMO systems such as UAVs, including the following:Stability: Adaptive backstepping control is a Lyapunov-based control method that ensures stability of the closed-loop system. This means that the control system is guaranteed to converge to a stable equilibrium point and remain there even in the presence of disturbances and uncertainties.Tracking performance: Adaptive backstepping control is capable of achieving high tracking performance, which is important for UAVs that need to follow specific flight paths and maintain a stable flight.Robustness: Backstepping control is a robust control method that can handle parameter uncertainties, external disturbances, and measurement noise. This is particularly important for UAVs that are subject to varying wind conditions, temperature changes, and other environmental factors that can affect their flight dynamics.Reduced chattering: Adaptive backstepping control produces control signals that are smooth and continuous, which reduces chattering in the control signal. This results in smoother control actions and reduces wear and tear on the UAV’s mechanical components.Reduced chattering: Backstepping control produces control signals that are smooth and continuous, which reduces chattering in the control signal. This results in smoother control actions and reduces wear and tear on the UAV’s mechanical components.Energy efficiency: Backstepping control can be designed to minimize energy consumption, which increases flight time and reduces the need for frequent battery replacements.Energy efficiency: Adaptive backstepping control can be designed to minimize energy consumption, which increases flight time and reduces the need for frequent battery replacements.Adaptability: Adaptive backstepping control is capable of adapting to changes in the UAV’s dynamics over time. This makes it a versatile control method that can handle different operating conditions and environmental factors.Easy implementation: Adaptive backstepping control can be implemented using standard digital signal processing techniques, making it easy to implement in modern UAV control systems.

The benefits of backstepping control make it a promising approach for controlling highly coupled systems such as UAVs. By improving the stability, tracking performance, robustness, energy efficiency, and ease of implementation of control systems, backstepping control can help to improve the performance and reliability of UAVs in a variety of applications.

#### Remarks

While backstepping and adaptive backstepping control have many benefits for controlling highly coupled systems such as UAVs, there are a number of limitations to this control method. The adaptive law used in adaptive backstepping control may not converge in certain situations, leading to unstable or oscillatory control performance. This can be a limitation for UAVs that operate in highly dynamic and uncertain environments. Backstepping and adaptive backstepping control are complex control methods that require a significant amount of computing power and may be difficult to implement in real-time applications. This can be a limitation for UAVs that require fast and accurate control responses. Backstepping and adaptive backstepping control require tuning of several parameters, such as the control gains and the stability functions. If these parameters are not tuned correctly, the control performance can be degraded. This can be a limitation for UAVs that operate in changing environments where the optimal parameter values may vary. Backstepping and adaptive backstepping control are dependent on the accuracy of the mathematical model used to describe the system dynamics. If the model is not accurate, the control performance can be compromised. This can be a limitation for UAVs that have complex and highly nonlinear dynamics. Backstepping and adaptive backstepping control require accurate measurements of the UAV’s state variables, such as its position and velocity. This can be a limitation for UAVs that have limited sensor capabilities or operate in environments where accurate measurements are difficult to obtain. Finally, designing a backstepping control for a complex UAV system can be challenging, and may require significant expertise in control theory and UAV dynamics.

Although backstepping and adaptive backstepping control have many benefits for controlling UAVs, these limitations must be carefully considered when selecting a control method for a particular application. Proper implementation and tuning are critical to ensure the control system’s stability, tracking performance, and robustness in the presence of disturbances and uncertainties.

### 6.4. Adaptive Backstepping Fast-Terminal Sliding Mode Control (AB-FTSMC)

Adaptive backstepping fast-terminal sliding mode control (ABFTSMC) is an advanced control strategy that has gained attention in recent years due to its ability to provide robust control for uncertain systems. In the context of a twin-rotor MIMO system, ABFTSMC can provide effective control for the system despite the presence of external disturbances and parametric uncertainties.

One of the advantages of ABFTSMC is that it combines the benefits of both adaptive control and sliding mode control. The adaptive control component allows the controller to adapt to changes in the system dynamics or uncertainties in real-time, while the sliding mode control component provides robustness against external disturbances and parametric uncertainties.

Recent studies have demonstrated the effectiveness of ABFTSMC for UAVs. For example, in [156], ABFTSMC was used to control a UAV with input nonlinearity. The authors demonstrated that the proposed controller can effectively mitigate the effects of input nonlinearity and external disturbances on the system. Similarly, in [157], ABFTSMC was used to control a twin rotor MIMO system with actuator saturation and external disturbances. The authors demonstrated the proposed controller’s ability to effectively attenuate the effects of actuator saturation and external disturbances and showed that it outperformed other control strategies.

#### Remarks

ABFTSMC uses adaptive backstepping and fast terminal sliding mode control techniques to achieve robust and accurate control of nonlinear MIMO systems. This controller is capable of handling uncertain system parameters, external disturbances, and input saturation, making it suitable for a wide range of practical applications. Simulation results [156,157] have demonstrated the effectiveness of the proposed controller in controlling nonlinear MIMO systems, with improved tracking performance and reduced control input chattering compared to other control methods. The proposed controller represents a significant contribution to the field of nonlinear control and has the potential to be applied in various real-world scenarios such as robotics, aerospace, and industrial control systems. ABFTSMC is highly robust against parametric uncertainties, external disturbances, and nonlinearities in the system. This makes it a suitable choice for controlling nonlinear MIMO systems, which are typically subject to these types of uncertainty. ABFTSMC incorporates adaptive control techniques, which means that it can adapt to changes in the system dynamics and uncertainties in real time. This makes it a highly flexible and adaptive control strategy. The fast=terminal sliding mode control component of ABFTSMC ensures that the control error converges to zero in a finite time. This means that the controller can quickly achieve the desired control objective even in the presence of uncertainties and disturbances. ABFTSMC can effectively handle input saturation, which is a common problem in MIMO systems. The adaptive component of the controller ensures that the system can operate within the input limits, while the sliding mode component ensures robustness against external disturbances and uncertainties.

### 6.5. Model Predictive-Based Sliding Mode Control (MPSMC)

Model predictive-based sliding mode control (MPSMC) is a hybrid control strategy that combines the benefits of MPC and SMC to control dynamic systems. In MPSMC, the SMC is used to design a sliding surface that ensures the system’s stability and tracking performance, while the MPC is used to optimize the control actions based on the predicted future behavior of the system.

The basic idea behind MPSMC is to use the SMC to design a sliding surface that guarantees the system’s stability and robustness against uncertainties and disturbances. The sliding surface is defined as a hyperplane that separates the system’s behavior into two regions: the sliding mode and the reaching mode. In the sliding mode, the system’s behavior is constrained to follow the sliding surface, while in the reaching mode the system’s behavior is guided towards the sliding surface. In [158], the authors proposed an MPSMC strategy for the stabilization and trajectory tracking of a UAV in the presence of uncertainties and disturbances, presented a detailed explanation of the MPSMC approach, and demonstrated its effectiveness through simulations. In [159], the MPSMC strategy was expressed with a different sliding surface design based on the predicted future behavior of the UAV system. The authors showed the effectiveness of their approach through simulations and experiments. In [160], the MPSMC approach was exploited for the attitude stabilization of a UAV using a quaternion-based representation of the system. The authors explained the details of their approach and demonstrate its effectiveness through simulations and experiments.

In their review article, the authors of [161] compared the performance of MPC, SMC, and MPSMC for the stabilization and trajectory tracking of UAVs. They provide a detailed explanation of the MPSMC approach and show its superiority in terms of tracking performance and robustness.

#### Remarks

After the sliding surface is designed, the MPC is used to optimize the control actions based on the predicted future behavior of the system. The MPC predicts the future states of the system and calculates the optimal control actions that minimize a cost function subject to constraints. The control actions are then applied to the system and the process is repeated at each time step. The MPC can handle constraints on the system’s inputs and states, ensuring that the control actions remain within physical limitations. MPSMC can be easily adapted to different systems and operating conditions, making it a versatile control method. The SMC guarantees the system’s stability and tracking performance, while the MPC optimizes the control actions to minimize a cost function subject to constraints. MPSMC is a promising approach for controlling dynamical systems, including highly coupled wind rotor MIMO systems such as UAVs, that require robustness, tracking performance, constraints handling, adaptability, and reduced chattering. By combining the benefits of MPC and SMC, MPSMC can help to improve the performance and reliability of control systems for a variety of applications. Despite its many benefits, there are limitations on the use of model predictive based sliding mode control (MPSMC) to control highly coupled wind rotor MIMO systems such as UAVs, which are discussed below.

The use of MPC in MPSMC can lead to increased computational complexity, especially for systems with large state and control spaces. This can lead to significant computational overhead and make it difficult to implement MPSMC in real-time control applications. MPSMC requires the tuning of several parameters, including the sliding surface, cost function, and prediction horizon, which can be challenging and time-consuming. In addition, the tuning process can be sensitive to changes in the system dynamics or environmental conditions, requiring frequent recalibration. MPSMC relies on accurate system models for prediction and optimization. However, there may be modeling errors due to uncertainties, disturbances, or unmodeled dynamics that can affect the accuracy of the control actions. The use of MPC in MPSMC can lead to input saturation, where the control inputs reach their physical limits. This can result in degraded control performance and reduced stability margins. The complexity of MPSMC can make it difficult to interpret and understand the control actions and their effects on the system. This can make it challenging to diagnose and correct control errors or to explain the system’s behavior to stakeholders.

### 6.6. Fuzzy Logic-Based Nonlinear Control Strategies

Instead of using the Boolean logic that computers typically utilize, fuzzy logic operates on the concept of “degree of truth”. Because the degree of truth can have any real value among 1 and 0, it is a many-valued logic. This control strategy’s logical thinking is similar to that of humans in that there are many options in between the Boolean values “yes” and “no”. Therefore, in this method the level of input possibilities is used to compute the precise output. The word “fuzzy” used to describe this control strategy alludes to its ability to produce predictable outputs for a variety of distorted and unclear inputs. Fuzzy also means uncertain or confused. While fuzzy logic controllers do not depend on a precise mathematical model, other linear and nonlinear controllers are more dependent on a precise and accurate mathematical model of the system. This characteristic, which relies solely on approximation reasoning, makes building a control system for any complicated system simpler. The min–max rules, which combine the membership functions using OR and AND logic, are a significant flaw in fuzzy logic [162,163]. Both the robustness and the human-like reasoning for several inputs are lacking from these principles. This control strategy has been applied in several UAV control investigations [162,163,164,165,166], and better outcomes have been noted [166,167]. Investigations in several articles [166,168,169] explain hybrid control strategies based on backstepping. A robust adaptive controller for a UAV was designed using the aforementioned control schemes after the UAV dynamical model was initially published. The adaptive fuzzy system was utilized to approximate the value of the generated control law, while backstepping was employed to derive the control law required to stabilize and track the intended location. The resulting controller assures asymptotic tracking. The developed adaptive fuzzy backstepping controller and the backstepping controller’s comparative findings were compared to alternatives, with the proposed controller performing accurate tracking [170].

Another study [171] concentrated on the notion of combining integral backstepping with fuzzy logic to enhance the performance of the flight controller. Simulation findings showed that the fuzzy-integral backstepping strategy performed better than traditional integral backstepping. An evolutionary approach was used to add a control structure to the fuzzy-based sliding mode control for UAVs. To prevent chattering, this method can be researched further using second-order sliding mode control.

#### Remarks

The use of fuzzy logic-based nonlinear control designs for UAV systems is appropriate in that a global controller can be created by locally defining control laws in terms of verbal expressions. Another benefit of this method is that no precise model is required for the design of the controller. Instead, it necessitates thorough knowledge and experience of the designer with the specific system. The system’s local behavior can be changed through the defined local rules, which are user-generated and depend on selected membership functions. While fuzzy controllers can technically be used with nonlinear systems, proving their stability and robustness analytically is time-consuming.

### 6.7. Neural Network-Based Nonlinear Control Strategies

Artificial neural networks (ANNs) process information similarly to organic neurons. In this paradigm, the structure of the linked neurons and their capacity for adaptive learning is crucial for information processing. The processing components of interconnected neurons cooperate to solve a particular issue and learn from examples in a manner similar to humans. Learning in ANNs occurs in a similar way to how it occurs in biological neurons when synaptic connections are changed by training. This learning can take place online or offline, allowing ANNs to be used in practical applications.

The utilization of ANNs for various tasks, such as data categorization, pattern identification, picture recognition, speech recognition, handwriting recognition, extracting patterns from large datasets, sales forecasting, and risk management, has been the focus of significant research efforts. Moreover, ANNs find applications in electronics and control systems. In the realm of unmanned aerial vehicles, several studies have employed ANNs to develop controllers for UAV models and understand the complex nonlinear dynamics through their approximation capabilities. A notable example is the comprehensive exploration of backstepping control and neural network adaptation conducted by Madani and Benallegue [100,172].

The deployment of ANNs spans a plethora of domains, including data categorization, pattern recognition, image analysis, speech processing, handwriting interpretation, data mining from extensive datasets, forecasting sales trends, and managing risks, all of which have garnered substantial attention in recent research endeavors. Furthermore, ANNs exhibit utility in electronics and control systems, showcasing considerable versatility. In the context of unmanned aerial vehicles, ANNs have been extensively utilized to craft controllers tailored to UAV models as well as to unravel the intricate nonlinear dynamics inherent in their operations, leveraging the approximation properties of neural networks. Noteworthy among these efforts is the in-depth investigation into backstepping control and neural network adaptation conducted by Madani and Benallegue [173].

The utilization of Artificial Neural Networks spans across various domains, encompassing tasks such as data classification, pattern recognition, image processing, speech analysis, and handwriting interpretation, as well as extracting insights from vast datasets for purposes such as sales prediction and risk assessment. Additionally, ANNs showcase their versatility in applications involving electronics and control systems. In the realm of unmanned aerial vehicles, ANNs have been extensively employed to develop controllers tailored to specific UAV models and to comprehend the intricate nonlinear dynamics governing their behavior, leveraging the approximation capabilities inherent in neural networks. A notable contribution in this regard is the exhaustive investigation into backstepping control and neural network adaptation undertaken by Madani and Benallegue [172].

One of the primary advantages of this proposed control strategy lies in its independence from detailed knowledge of physical characteristics and system models. Operating under a few broad assumptions, this control method ensures the convergence of all closed-loop dynamic states. Furthermore, it offers versatility, as it can be applied to UAVs of varying masses and inertia within the same class without necessitating knowledge of specific physical parameters. Notably, the concept of online adjustment of AANN weights, obviating the need for a separate learning phase, is elucidated in this approach. Nonlinearities inherent in the UAV model are estimated through multilayer ANNs.

Several studies [110,174,175] have explored dynamic inversion and direct inverse control methods incorporating ANNs within the control system architecture. Meanwhile, sliding mode control stands out as a prevalent technique for managing nonlinear systems. This method operates by introducing a sliding manifold, which is a low-dimensional subspace of the system’s state space along which the system’s state is steered towards a desired trajectory.

In [176], the authors proposed a neural network-based adaptive backstepping control approach for nonlinear MIMO systems with input saturation. The proposed approach uses neural networks to approximate the unknown nonlinearities of the system and adaptive laws to estimate the saturation boundaries of the input signals. The backstepping design approach is employed to design a series of feedback control laws that drive the system’s state to a desired trajectory. The control approach was shown to be effective in controlling a two-link robot arm system, and the results were compared with those obtained using a traditional backstepping control approach. The proposed approach was demonstrated to outperform the traditional approach in terms of tracking accuracy, control effort, and robustness against disturbances and uncertainties. This article provides a useful contribution to the field of adaptive backstepping control and its application to nonlinear MIMO systems with input saturation. Another article [177] proposed an adaptive neural network backstepping control approach for MIMO nonlinear systems with time-varying input delay. The proposed approach uses a neural network to approximate the unknown nonlinearities of the system and a backstepping design approach to design a series of feedback control laws that drive the system’s state to a desired trajectory. An adaptive law was developed to estimate the time-varying input delay, which was then compensated for in the control design. The control approach was shown to be effective in controlling a three-tank system with time-varying input delay, and the results were compared with those obtained using a traditional backstepping control approach. The proposed approach was demonstrated to outperform the traditional approach in terms of tracking accuracy and robustness against input delay variations. This article provides a useful contribution to the field of adaptive backstepping control and its application to MIMO nonlinear systems with time-varying input delay.

In [178], the authors proposed a robust adaptive neural network backstepping control approach for MIMO nonlinear systems with input saturation. The proposed approach uses a neural network to approximate the unknown nonlinearities of the system and a backstepping design approach to design a series of feedback control laws that drive the system’s state to a desired trajectory. An adaptive law was developed to estimate the saturation boundaries of the input signals, which were then compensated for in the control design. The control approach was shown to be effective in controlling an unmanned aerial vehicle system with input saturation, and the results were compared with those obtained using a traditional backstepping control approach. The proposed approach was demonstrated to outperform the traditional approach in terms of tracking accuracy and robustness to input saturation. This article provides a useful contribution to the field of adaptive backstepping control and its application to MIMO nonlinear systems with input saturation. Learning the nonlinear behavior of any complex system using neural network approximation property is a worthwhile technique to account for the system’s uncertainties and bounded disturbances. The nonlinear functions [179,180,181] have been commonly approximated using radial basis functions (RBFs). Extended radial basis functions (E-RBFs) and normalized radial basis functions (*N*-RBFs) are examples of augmented RBF functions. In [181], both *N*-RBFs and E-RBFs were used to approximate a UAV’s unknown dynamics.

In a few studies, the position controller has been viewed as a MIMO nonlinear system and the function of the system approximated using a radial basis function neural network (RBFNN). This is described in [182,183]. The originality of this study is in the online training of an adaptive rule to approximate a complete controller. This controller, which is RBFNN-based, offers adaptation for parametric uncertainties as well as other outside disturbances. The controller’s generated outputs include the total thrust term as well as the necessary roll and pitch angles for the attitude subsystem. Reinforcement learning (RL) has been used to successfully tackle many complex issues, and well-trained networks have surpassed human specialists in many challenging applications. Unfortunately, the majority of robotics research on reinforcement learning has mostly been limited to higher-level choices, while the state-actuator space, e.g., low-level actuator commands, has not received much attention. A UAV flight controller was trained using reinforcement learning in [183,184]. In [185], the authors proposed a neural network-based SMC method for a UAV. This control algorithm combines the advantages of SMC and neural network control to achieve improved control performance. Another paper [186] presented a neural network-based SMC method for a UAV. The proposed control approach is designed to improve the tracking performance of the system while maintaining robustness against parameter uncertainties and external disturbances. A neural network-based SMC approach for a UAV was presented in [187]. The control algorithm employs a feedforward neural network to approximate the unknown system dynamics and an SMC approach to handle parameter uncertainties and external disturbances.

Another neural network-based SMC method for a UAV was presented in [188]. The control approach employs a feed-forward neural network to approximate the system dynamics and a SMC approach to handle parameter uncertainties and external disturbances. The proposed control approach was tested on a real-time experimental platform, and the experimental results demonstrated its effectiveness in controlling the UAV. In [184], a policy network was trained to use high-quality deterministic samples to translate UAV states into thrust commands. While the majority of recent studies have provided learning rules based on stochastic samples, deterministic policies use comparatively fewer samples than stochastic policies, which lowers the processing cost. In order to control the flight of a UAV, a model-based reinforcement learning policy was implemented in [189]. However, the synthesized controller does not exhibit a promising response to the step input. In [190], a neural network-based adaptive control was developed for UAV formation flight.

A small amount of research on UAV model neuro-fuzzy controller systems has been published as well. Through the use of an adaptive neuro-fuzzy inference system, Bhatkhande and Havens created an intelligent neuro-fuzzy controller [163]. The developed controller was trained using traditional PD controller data. The controller’s real-time implementation demonstrated acceptable control performance. Similar outcomes were reported in [191]. A UAV attitude controller was developed using type-2 fuzzy neural networks according to an advanced study [192,193]. In this study, the network-based controller was trained using the best parameter update rule. Unmanned aerial vehicle safety is crucial, particularly when they are utilized for military operations and public missions. The UAVs must be sophisticated enough to successfully mitigate any potential undesirable condition in the event of any introduced defect, whether hardware- or software-based [163,191,193]. Insufficient research has been carried out on fault-tolerant control of quadrotors, and there are a number of related problems that need to be addressed thoroughly, such as preventing cyberattacks during high-profile spying missions, identifying injected flawed data, and detecting sensor spoofing and wireless communication attacks. The authors of [190,193] presented a thorough analysis of the susceptibility of autopilots to potential cyberthreats and discussed several cyberattack tactics.

#### Remarks

Ensuring resilience and tolerance against cyberthreats such as injected false data, wireless communication attacks, and sensor spoofing, is imperative for intelligent control of UAVs. Several research teams have delved into the utilization of neural network-based observers to address the aforementioned challenges. However, this underscores the pressing need for further research aimed at developing control structures that are resistant to cyberattacks and other potential UAV hijacking scenarios. A fault-tolerant flight controller must possess the ability to keenly sense anomalies and exhibit low susceptibility to disturbances. Moreover, it should be capable of accurately estimating the magnitude and severity of faults as well as promptly and precisely detecting and diagnosing them.

In the realm of UAV technology, sensor-based adaptive controls are revolutionizing capabilities, facilitating more intelligent, autonomous, and efficient operations [78]. These systems play a pivotal role in broadening the scope of applications for UAVs and represent a crucial area of focus in both current and future UAV technology development. An overview of sensor-based adaptive strategies are elaborated in Figure 13.

### 6.8. Integration of Sensors into UAV Control Systems

The integration of sensors into UAV control systems is a critical aspect of their operation, enhancing their capabilities and allowing for a wide range of functionalities [79,80]. This process involves several key elements and considerations.

Types of Sensors Used in UAVs

Navigation Sensors: GPS for location, gyroscopes for orientation and balance, accelerometers for speed and direction, magnetometers for heading.Environmental Sensors: Lidar for terrain mapping, infrared and thermal sensors for night vision or heat mapping, multispectral cameras for environmental monitoring.Obstacle Detection Sensors: Ultrasonic sensors, radars, optical cameras for real-time obstacle avoidance and situational awareness.

#### Sensor Fusion

Combining Data: Sensor fusion involves integrating data from multiple sources to create a more comprehensive understanding of the UAV’s environment. For instance, combining GPS and IMU data can provide accurate positioning and movement information [81].Enhanced Decision-Making: Sensor fusion allows for more precise control and decision-making, as it mitigates the limitations of individual sensors and leverages their strengths [82,83].


**Control System Integration**


Feedback Loops: Sensors provide critical real-time data that feeds into the UAV’s control system, enabling it to adjust its flight path, speed, altitude, and orientation [92].Autonomous Operations: For autonomous UAVs, sensors are integral to their ability to navigate, perform tasks, and make decisions independently [92,93].


**Data Processing and Analysis**


Onboard Processing: Advanced UAVs often have onboard computers to process sensor data in real time, enabling immediate response to environmental changes.Algorithm Development: Developing algorithms that can efficiently and accurately process sensor data is crucial. These algorithms must be capable of handling high volumes of data from various sensors simultaneously.


**Calibration and Synchronization**


Sensor Calibration: Calibration ensures that sensors are accurately calibrated, which is vital for obtaining reliable data. Incorrect calibration can lead to errors in navigation and environmental interpretation [51].Time Synchronization: Synchronizing data from various sensors is essential, especially when combining data for decision-making [101].


**Energy and Resource Management**


Power Consumption: Sensors and data processing both consume power. Balancing energy consumption with operational efficiency is crucial, especially for missions requiring longer flight times [102].Weight and Space Constraints: The size and weight of sensors need to be considered, as they affect the UAV’s payload capacity and flight dynamics [103].


**Challenges and Considerations**


Environmental Factors: Sensors must be robust enough to operate in various environmental conditions, including weather changes, lighting variations, and temperature extremes [123].Reliability and Redundancy: Ensuring sensor reliability and incorporating redundancy for critical sensors is important for the safe operation of UAVs [123,124].

The integration of sensors into UAV control systems is a multifaceted process that enhances UAVs’ capabilities in navigation, task execution, and environmental interaction. It involves careful consideration of sensor selection, data fusion, control integration, and resource management to ensure efficient and effective UAV operations.

### 6.9. How Sensor Data Enhance UAV Control Strategies

Sensor data play a pivotal role in enhancing the control strategies of Unmanned Aerial Vehicles (UAVs), enabling more sophisticated, reliable, and adaptive operations [123,125]. Sensor data contribute to this enhancement in several ways.


**Improved Situational Awareness**


Comprehensive Environmental Understanding: Sensors such as cameras, Lidar, and radar provide UAVs with a detailed understanding of their surroundings. These data is crucial for navigation, obstacle avoidance, and mission execution in complex environments.Real-Time Adjustments: With real-time data, UAVs can adapt to changing environmental conditions such as weather changes, unforeseen obstacles, and variable terrain, thereby improving flight safety and effectiveness.


**Enhanced Navigation and Positioning**


Precise Location Tracking: GPS and IMU data enable accurate positioning, which is essential for waypoint navigation and geospatial tasks [126].Stable Flight Control: Gyroscopes and accelerometers provide critical information about the UAV’s orientation and movement, enabling stabilization and precise maneuvering [127].


**Optimized Flight Paths and Energy Efficiency**


Efficient Route Planning: Sensor data can be used to optimize flight paths for energy efficiency, reduce battery usage, and extend mission duration [135].Adaptive Speed Control: By analyzing environmental data, UAVs can adjust their speed to conserve energy or avoid hazards, contributing to smarter energy management [136].


**Advanced Task Execution**


Target Detection and Analysis: Specialized sensors such as thermal or multispectral cameras allow UAVs to perform specific tasks such as crop monitoring, search and rescue, and infrastructure inspection with greater accuracy [136].Automated Payload Deployment: In applications such as agricultural spraying or package delivery, sensor data can guide precise payload deployment, enhancing the effectiveness of these operations [137].


**Improved Safety and Collision Avoidance**


Obstacle Detection: Ultrasonic sensors, Lidar, and optical cameras enable UAVs to detect and avoid obstacles, which is crucial for safe operation in crowded or dynamic spaces [138].Emergency Response: Sensor data can trigger automatic safety protocols such as return-to-home or landing procedures in response to critical situations such as battery failure or extreme weather conditions [194].


**Facilitation of Autonomous Operations**


Self-Guided Systems: Integration of sensor data is essential for fully autonomous UAVs. These data allow UAVs to make independent decisions about navigation and task execution and to respond to environmental changes [143].Machine Learning and AI Integration: Sensor data can be fed to machine learning algorithms, enabling UAVs to learn from past experiences, improve their responses, and handle complex tasks with greater autonomy [144].


**Enhanced Communication and Data Transmission**


Data for Ground Control: Sensor data transmitted to ground control stations provide operators with essential information for remote decision-making and intervention when necessary [145].Network Integration: In applications involving IoT, sensor data can be used to integrate UAVs into broader networks, facilitating tasks such as data collection and monitoring across various locations [147].

Sensor data can significantly enhance UAV control strategies by providing the necessary inputs for improved situational awareness, precise navigation, energy-efficient operations, advanced task execution, safety protocols, autonomous functionalities, and effective communication. This integration of sensor data into UAV systems is fundamental to the advancement and diversification of UAV applications.

### 6.10. Examples of Sensor-Based Adaptive Control Methods in UAVs

Sensor-based adaptive control methods in Unmanned Aerial Vehicles (UAVs) leverage real-time data from various onboard sensors to dynamically adjust control strategies [148]. These methods enhance the ability of UAVs to operate effectively and execute complex tasks in varying conditions.


**GPS-Assisted Navigation**


Waypoint Navigation: GPS data are used to guide the UAV along predefined coordinates while adapting its path based on real-time location data [149].Geofencing: GPS sensors enable UAVs to recognize and adhere to virtual boundaries, automatically adjusting their flight path to stay within designated areas [151].


**Vision-Based Obstacle Avoidance**


Optical Flow Sensors: When combined with cameras, these sensors allow UAVs to detect and avoid obstacles by analyzing visual data and adapting their flight path accordingly [152].Stereo Vision: By using two cameras to simulate 3D vision, UAVs can gauge the distance and size of obstacles, then adjust their flight to avoid collisions [153].Lidar for Terrain Mapping: UAVs equipped with Lidar sensors can create detailed 3D maps of terrain and structures, helping to adapt their altitude and position for precise mapping [155].


**Inertial Measurement Unit (IMU) for Stabilization**


Gyroscopic Control: Gyroscopes in the IMU provide data for roll, pitch, and yaw stabilization, adapting control inputs to maintain stable flight [156].Accelerometer Data: Accelerometers aid in maintaining a steady altitude and velocity and in adjusting the UAV’s thrust and tilt in response to changes in movement [157].Thermal Imaging for Search and Rescue: UAVs can use thermal sensors to locate people or animals by their heat signatures, especially in low-visibility conditions, by adapting their search patterns based on thermal data [158].Multispectral Imaging for Precision Agriculture: These sensors enable UAVs to monitor crop health by capturing data in various spectral bands by adapting their flight over farmlands to identify areas needing attention [159].Wind Sensors for Energy-Efficient Flight: By measuring wind speed and direction, UAVs can adapt their flight path and speed for more energy-efficient routing [162,163].Ultrasonic Sensors for Indoor Navigation: In indoor environments or when flying close to surfaces, ultrasonic sensors can help maintain a safe distance by adapting the UAV’s altitude and position to avoid collisions [163].

These sensor-based adaptive control methods demonstrate the enhanced capabilities of modern UAVs, enabling them to perform a wide range of tasks more safely and effectively. The integration of sensor data and intelligent utilization is fundamental in enhancing UAV adaptability and potential applications.

## 7. Importance of Artificial Intelligence Control Techniques

Hybrid control theory is a specialized area within modern control theory that focuses on systems demonstrating a combination of continuous and discrete behaviors. These systems, referred to as hybrid systems, are marked by the interaction between continuous dynamics and discrete events. Continuous dynamics refer to gradual changes in the system’s state over time, much like those seen in traditional control systems, where variables evolve smoothly; in contrast, discrete events represent abrupt changes or transitions in the system’s behavior, such as switching between different operational modes or the occurrence of events triggering sudden alterations in system dynamics [141].

For instance, consider an autonomous vehicle system. The continuous dynamics might govern factors such as vehicle speed and acceleration, which change smoothly over time. However, discrete events could include actions such as switching between driving modes (e.g., from manual to autonomous) or reacting to sudden obstacles on the road [142].

Hybrid control theory addresses the unique challenges posed by systems exhibiting both continuous and discrete behaviors. It focuses on understanding and effectively managing the interaction between these dynamics, enabling the design of control strategies that can handle the complexities inherent in hybrid systems [136,195].

### Real Word Applications of Control Strategies

In addition to theoretical advancements, the practical implementation of advanced control strategies has demonstrated significant improvements in UAV operations across various real-world applications. For instance, MPC has been successfully applied in autonomous surveillance missions, enabling UAVs to dynamically track and follow moving targets while avoiding obstacles in complex urban environments [196]. Feedback linearization control has proven effective in enhancing the stability and maneuverability of UAVs during high-speed flight operations, making it suitable for applications such as search and rescue missions or aerial photography [197]. Moreover, the integration of sensor-based adaptive controls and artificial intelligence-driven approaches has enabled UAVs to adapt and respond to changing environmental conditions, enhancing their versatility and ability to perform a wide range of tasks [198]. These case studies and real-world applications highlight the practical impact and potential of advanced control strategies in advancing UAV capabilities and expanding their range of applications [199]. By leveraging the insights and methodologies discussed in this review, researchers, engineers, and practitioners can further enhance the performance, autonomy, and reliability of UAV systems [200].

## 8. Recent Developments and Research Directions in UAV Control Strategies

In recent years, the field of UAV control has seen significant advancements driven by technological innovations and the increasing complexity of applications [171].

### 8.1. Key Developments in UAV Control Strategies

Advanced Sensor Integration: Enhanced use of sensors such as Lidar, GPS, thermal imaging, and multispectral cameras has improved UAV capabilities in navigation, obstacle avoidance, and task-specific operations [171].Improved Sensor Fusion Techniques: Developments in sensor fusion have allowed for more accurate and reliable interpretation of environmental data, enhancing situational awareness and decision-making [173,201].AI and Machine Learning Integration: The incorporation of AI and machine learning has led to smarter UAVs capable of adaptive decision-making, predictive analytics, and learning from past operations [172].Nonlinear and Adaptive Control Systems: To address the challenges posed by nonlinear dynamics and coupling effects in UAVs, recent control strategies have focused on adaptive and robust control systems that can operate effectively in a wide range of conditions [176].Autonomous and Collaborative Operations: Progress in autonomous control has enabled UAVs to perform complex tasks with minimal human intervention. Developments in swarm technology have opened avenues for collaborative UAV operations [177].

### 8.2. Current Research Trends and Emerging Techniques

Energy-Efficient Control Algorithms: With the growing need for longer flight times, research is focusing on developing control strategies that optimize energy consumption [132].Enhanced Autonomy in Unstructured Environments: Researchers are exploring ways to improve UAV autonomy in complex and unstructured environments, such as dense urban areas or natural disaster sites [178].Human–Machine Interaction: There is increasing interest in intuitive control interfaces and systems that allow seamless human–UAV interaction, especially for applications such as search and rescue and surveillance [131,132].5G and IoT Integration: Leveraging 5G networks and IoT integration for real-time data transmission and enhanced connectivity is a growing research area [130,131].Swarm Intelligence: The study of UAV swarms for coordinated multi-UAV operations offers significant potential for applications such as agricultural monitoring, environmental surveillance, and defense [128,130,131,132].

The advancements in UAV control strategies and the current focus of research reflect the evolving needs and potentials of UAV technology. These developments promise to enhance the capabilities, efficiency, and applicability of UAVs in various sectors.

### 8.3. Challenges and Opportunities in Current Research

While there are exciting prospects in UAV research, several challenges and opportunities need to be addressed:Regulatory and Safety Concerns: Ensuring that UAV operations comply with evolving regulatory frameworks and addressing safety concerns in shared airspace is a major challenge.Cybersecurity: As UAVs become more connected and autonomous, they face increasing risks due to cyberthreats. Research into secure communication and data transmission is crucial.Robustness in Diverse Conditions: Developing UAV control systems that are robust in a variety of environmental conditions, including adverse weather or GPS-denied environments, remains a challenge.Scalability of Swarm Technologies: While swarm technology is promising, scaling it for large-scale operations poses technical and logistical challenges.Ethical and Privacy Considerations: As UAVs become more pervasive, addressing ethical and privacy concerns is essential, especially in surveillance applications [194].

The field of UAV control is rapidly evolving, with significant advancements in sensor technology, AI, and autonomous systems driving research forward. Addressing the challenges in safety, regulatory compliance, cybersecurity, and ethical considerations will be key to unlocking the full potential of these technological developments.

## 9. Future Challenges and Opportunities in UAV Control Strategies

Greater Autonomy and Intelligence: Future UAV control systems are expected to exhibit higher levels of autonomy and intelligence and be capable of complex decision-making with minimal human input. Developments in AI and machine learning will enable UAVs to learn from experiences and adapt to new situations more effectively [140].Advanced Swarm Coordination: There is potential for significant advancements in swarm coordination, allowing for more complex and scalable UAV operations. This includes enhanced communication systems and algorithms for real-time adaptive coordination among multiple UAVs [135].Robustness in Adverse Conditions: Control strategies that maintain robustness in challenging environments such as extreme weather, high electromagnetic interference, or GPS-denied areas are likely to be a key focus [134,135].Energy-Efficient Designs: With the increasing emphasis on sustainability, future control strategies will need to prioritize energy efficiency, including the development of algorithms that minimize power consumption and extend flight durations [128,130,131,132].

Environmental and Wildlife Monitoring: Advanced UAVs can play a crucial role in large-scale environmental monitoring, biodiversity conservation, and management of natural resources, especially in remote and inaccessible areas [123,125].Urban Planning and Smart Cities: UAVs equipped with advanced control systems will be instrumental in urban planning, traffic management, and infrastructure maintenance within smart cities [123].Disaster Response and Humanitarian Aid: UAVs can provide rapid real-time information and aid delivery in disaster-stricken areas, significantly improving emergency response efforts [125,127].Healthcare and Medical Delivery: There is potential for UAVs to be used in remote or urgent medical deliveries such as transporting medication, blood, or medical supplies to hard-to-reach areas [127].Space Exploration: Advanced control systems could enable UAVs to explore extraterrestrial environments, such as Mars or other planets, where manual control is not feasible [125].

### Significance of Sensor Integration in UAVs

Sensor integration plays a pivotal role in advancing the capabilities and performance of UAV control systems. Recent advancements in sensor technology have enabled significant improvements in UAV navigation, stability, and autonomous operation [202]. For instance, the integration of high-precision Global Navigation Satellite Systems (GNSS) with inertial sensors has enhanced UAV positioning accuracy, enabling precise navigation and waypoint following even in GPS-denied environments. Additionally, the adoption of advanced optical and infrared cameras along with LiDAR and radar sensors has revolutionized the sensing capabilities of UAVs for obstacle detection and avoidance. Furthermore, the incorporation of sophisticated environmental sensors such as air quality and temperature sensors has expanded the scope of UAV applications to include environmental monitoring and surveillance. These examples illustrate the practical impact of sensor integration on UAV control systems, demonstrating how cutting-edge sensor technologies are driving advancements in UAV performance, reliability, and autonomy. By leveraging these integrated sensor systems, UAVs are capable of executing complex missions with increased efficiency and safety, paving the way for broader adoption in various industries and applications [196,203,204].

## 10. Open Research Questions and Future Research Directions

Human–UAV Interaction: How can interfaces for human–UAV interaction be improved to make them more intuitive and effective, especially for untrained users?Ethical and Privacy Concerns: What are the ethical implications of widespread UAV usage and how can privacy concerns be addressed, especially in surveillance applications?Integration with Manned Aircraft: How can UAVs be safely and effectively integrated into existing airspace, which is predominantly occupied by manned aircraft?Counter-UAV Systems: As UAVs become more common, what are the strategies for counter-UAV systems to prevent misuse or hostile UAV activities?Cross-Domain Coordination: What are the prospects and challenges around coordinating UAV operations across different domains, such as air, ground, and maritime?

The future of UAV control strategies is poised for significant advancements, offering a myriad of opportunities across various sectors. Addressing open research questions and overcoming challenges in integration, safety, and ethics will be crucial in realizing the full potential of UAV technology in the years to come.

## 11. Conclusions

In this survey, we have thoroughly examined UAV control systems, focusing on TRMS and quadrotor platforms, and have highlighted the necessity of advanced nonlinear control strategies due to system complexity. While sensor integration enhances UAV capabilities, it introduces challenges in data management. Traditional linear methods fall short, while sensor-based adaptive and AI-driven approaches excel. Looking ahead, ongoing advancements will shape UAV capabilities, unlocking new opportunities in complex environments. This survey bridges theory and practice, deepening the understanding of nonlinear control and the pivotal role of sensor integration in advancing UAV capabilities.

## Figures and Tables

**Figure 1 sensors-24-03286-f001:**
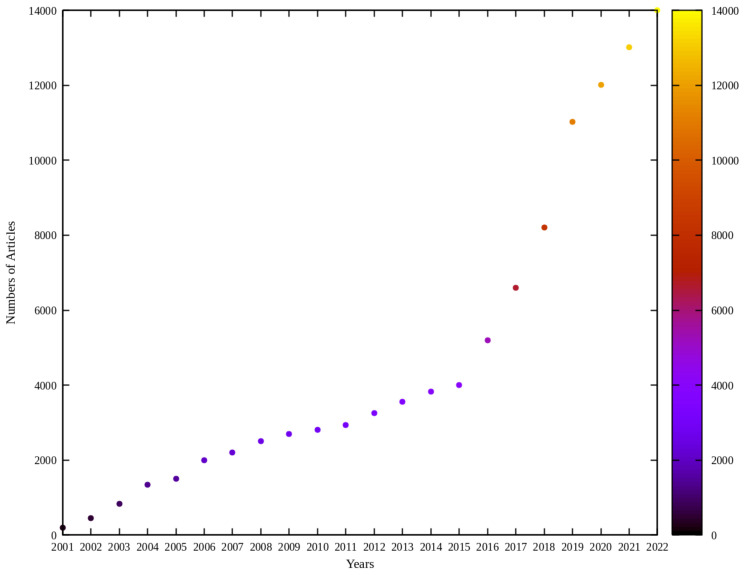
Scientific research articles according to SCOPUS [10].

**Figure 2 sensors-24-03286-f002:**
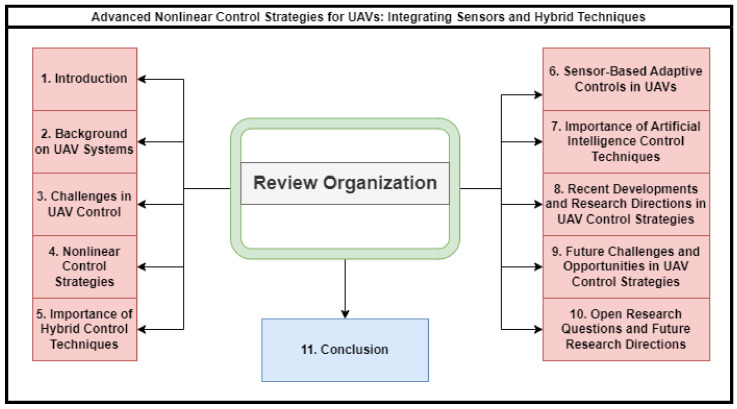
Survey article organization.

**Figure 3 sensors-24-03286-f003:**
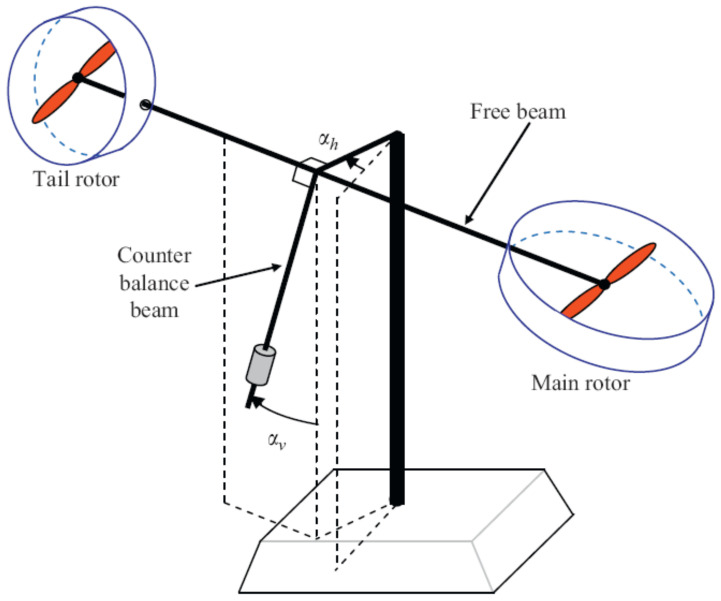
Twin-rotor aerodynamic system [6].

**Figure 4 sensors-24-03286-f004:**
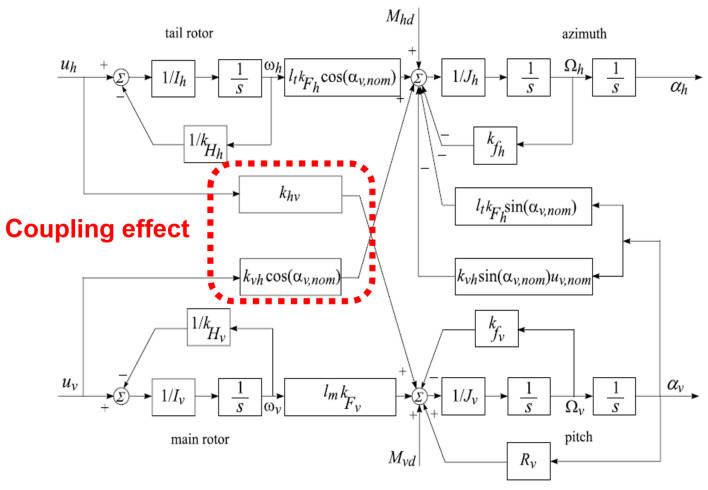
Mathematical model-based block diagram of UAV with coupling effect.

**Figure 5 sensors-24-03286-f005:**
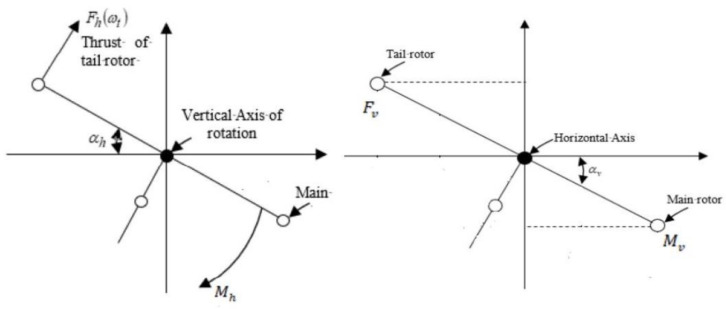
Rotor dimensions with torque.

**Figure 6 sensors-24-03286-f006:**
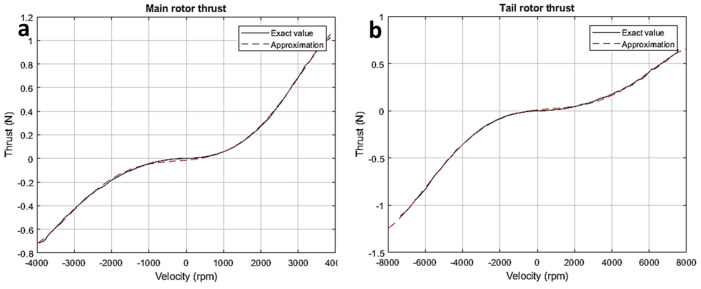
(**a**) Main rotor thrust and (**b**) tail rotor thrust.

**Figure 7 sensors-24-03286-f007:**
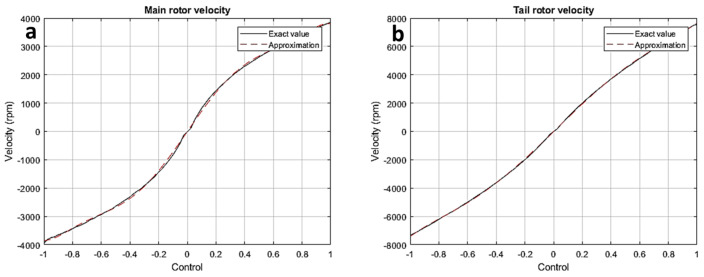
(**a**) Main rotor velocity and (**b**) tail rotor velocity.

**Figure 8 sensors-24-03286-f008:**
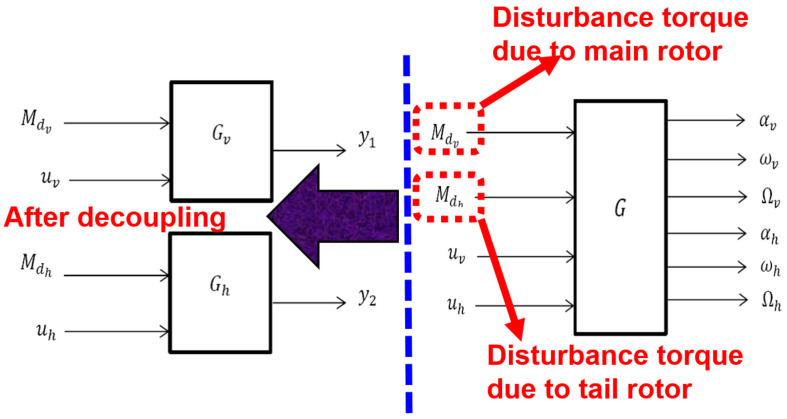
Block diagram of decoupled UAV input/output.

**Figure 9 sensors-24-03286-f009:**
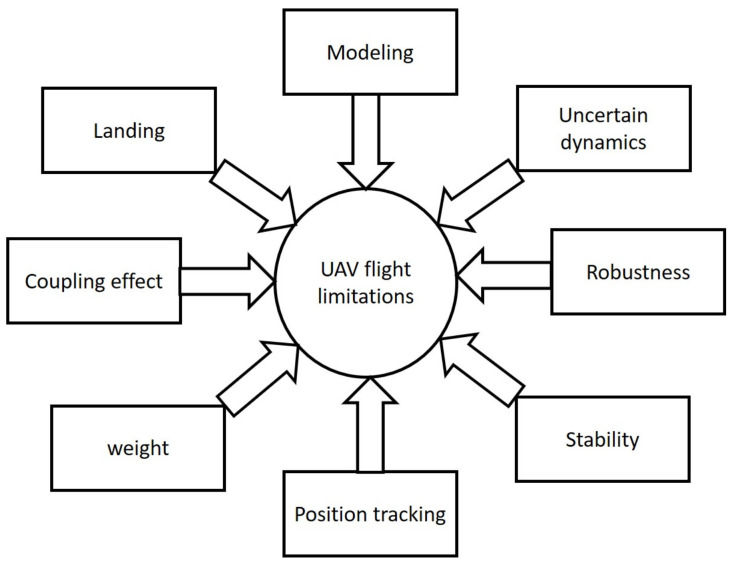
Challenges in UAV control.

**Figure 10 sensors-24-03286-f010:**
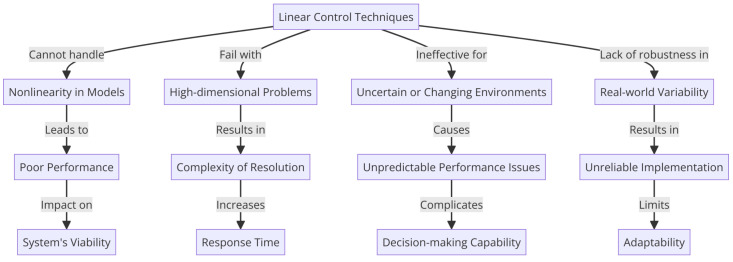
Limitations of traditional linear control techniques.

**Figure 11 sensors-24-03286-f011:**
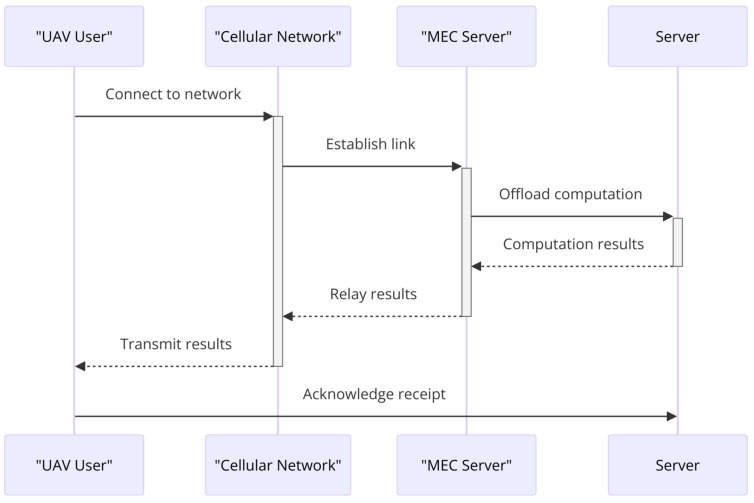
Sensor data complexity and UAV control.

**Figure 12 sensors-24-03286-f012:**
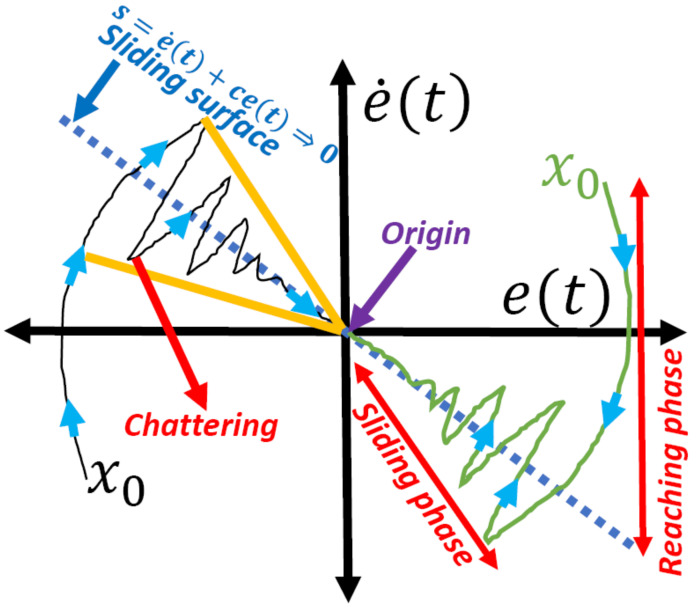
Chattering phenomenon.

**Figure 13 sensors-24-03286-f013:**
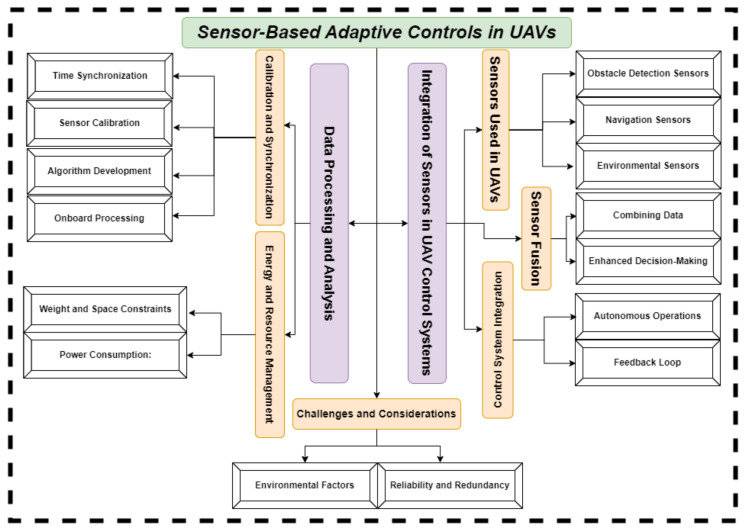
Overview of sensor-based adaptive control of UAVs.

**Table 1 sensors-24-03286-t001:** Overview of recent surveys on UAV communication and control strategies.

Authors, Year	Reference	Brief Description
Fotouhi, et al.	[13], 2020	Review providing an exhaustive perspective on the use of UAVs and discussing practicalities including the hurdles of integration, creation of protocols, establishment of standards, and issues pertaining to security.
Wang, et al.	[14], 2022	Analysis delving into progress on UAVs encompassing essential principles, application contexts, Air-to-Ground (A2G) communication channels, and the functions of UAVs. It includes an evaluation of secrecy performance and improvement strategies for both stationary and mobile UAV systems.
Han	[15], 2023	Survey offering a comprehensive examination of studies regarding the utilization and path planning of Unmanned Aerial Vehicles (UAVs) to improve the capacity and management of UAV wireless networks. It also underscores the hurdles faced in this field and suggests potential directions for future research.
Sharma, et al.	[16], 2021	Review centering on cutting-edge network technologies for UAVs and their deployment in upcoming cellular networks, exploring a range of nascent communication technologies for UAVs and evaluating their benefits, prospective uses, technical hurdles, and prospective developments.
Hentati, et al.	[17], 2021	Examination providing an in-depth analysis of UAV communication protocols, network architectures, frameworks, and practical applications, while emphasizing significant technical obstacles and identifying critical areas of research that demand further exploration and advancement.
Xiao, et al.	[18], 2022	Review presenting a comprehensive summary of research pertaining to UAV communications and the integration of technologies. It explores the domain of mmWave beamforming in UAV communications, addressing the technical potential and difficulties, and delves into the pertinent aspects of mmWave antenna design and channel modeling.
Geraci, et al.	[19], 2023	A study demonstrating the efficacy of sub-6GHz massive MIMO technology in handling cell selection and interference, evaluating the coverage of mmWave frequencies in various environments, and scrutinizing the intricacies of initial 2D communication for airborne devices.
McEnroe, et al.	[20], 2023	Review investigating how edge artificial intelligence influences crucial technical aspects and applications of UAVs, spanning domains such as power management, formation control, autonomous navigation, and computer vision while addressing concerns related to privacy, security, and communication.
Jasim, et al.	[21], 2022	Review identifying appropriate management strategies for UAV characteristics and spectrum requirements, taking into account their coexistence with current wireless technologies within the spectrum. Additionally, it details the guidelines and directives of policymakers and regulators and investigates various operational frequency bands and radio interfaces.
Xu, et al.	[22], 2022	Review scrutinizing the evolution of regulatory policies and key technologies pertinent to the safe and effective functioning of small civilian UAVs operating at low altitudes in urban settings.
Hafeez, et al.	[23], 2023	Review focusing on integrating privacy and security measures in blockchain-supported UAV communications, underscoring the need for basic analysis and decentralized data storage solutions while laying out crucial prerequisites in the formulation of privacy and security frameworks.
Wei, et al.	[24], 2023	An assessment offering an extensive examination of various scenarios and pivotal technologies relevant to UAV-assisted data gathering in the Internet of Things (IoT) context. It outlines system architectures, encompassing both the network infrastructure and mathematical modeling, and performs an in-depth evaluation of the essential technologies involved.
Nomikos, et al.	[13], 2023	Review investigating the role of UAVs in maritime communications and the integration of conventional approaches with machine learning techniques to improve performance in aspects such as the physical layer, resource allocation, and cloud/edge computing.
Duong, et al.	[25], 2023	Review presenting an exhaustive overview of UAV caching within 6G networks, covering the progression of caching models from ground-based to aerial systems. It introduces a standard UAV caching system and delves into the latest developments and performance indicators in this field.
This Survey		A survey reviewing advanced nonlinear control strategies for UAVs, emphasizing the necessity of sensor-based adaptive controls and artificial intelligence. It explores innovative control strategies such as sliding surface and sensor-driven techniques, highlighting their effectiveness in enhancing UAV performance and stability. This review underscores the complexities of UAV control, the critical role of sensors, and the benefits of nonlinear methods while discussing recent advancements and future challenges in this rapidly evolving field.

**Table 2 sensors-24-03286-t002:** Parameters of TRMS (UAV).

Variable Notation	Description	Units and Values of Parameters
$I_{1}$	Main rotor inertia	$6.8\times 10^{-2}$ kgm^2^
$I_{2}$	Tail rotor inertia	$2\times 10^{-2}$ kgm^2^
$a_{1}$	constant	0.0135
$b_{1}$	constant	0.0924
$a_{2}$	constant	0.02
$b_{2}$	constant	0.9
$M_{g}$	Gravitational Momentum	0.32 Nm
$B_{1\theta }$	Frictional parameter	$6\times 10^{-3}$ Nms^2^/rad^2^
$B_{2\theta }$	Frictional parameter	$1\times 10^{-3}$ Nms^2^/rad^2^
$B_{1\varphi }$	Frictional parameter	$1\times 10^{-1}$ Nm.s/rad
$B_{2\varphi }$	Frictional parameter	$1\times 10^{-2}$ Nms^2^/rad
$k_{gy}$	Gyroscopic Parameter	0.05 rad/s
$k_{1}$	Gain of Main Motor	1.1
$k_{2}$	Gain of Tail Motor	0.8
$T_{11}$	Denominator of motor	1.1
$T_{10}$	Numerator of motor	1
$T_{21}$	Denominator of motor	1
$T_{20}$	Numerator of motor	1
$k_{c}$	Coupling reaction for gain	2

**Table 3 sensors-24-03286-t003:** Hybrid control developments.

Year	Contributor	History
1980s	Edward A. Lee and Alberto L. Sangiovanni-Vincentelli	Proposed a framework for analyzing the behavior of hybrid systems, introducing the concept itself.
1990s	Rajeev Alur, Thomas A. Henzinger, and Orna Kupferman.	Over time, hybrid control theory has witnessed significant progress, propelled by contributions from numerous researchers. Among the pivotal advancements in this domain is the introduction of the hybrid automaton model. This model serves as a formal framework for both representing and scrutinizing the intricate dynamics of hybrid systems [136,137].
1995	Henzinger, T. A., and Kopke, P. W.	Additional significant contributions to hybrid control theory encompass the formulation of control synthesis methodologies. These methods are instrumental in crafting effective control strategies tailored specifically for hybrid systems. Moreover, extensive research has been dedicated to investigating the stability and performance characteristics inherent in hybrid systems, further enriching the understanding of their complex behavior [138].
2008	Cassandras, C. G., and Lafortune, S.	Another noteworthy advancement in hybrid control theory is the emergence of reachability analysis techniques. These methods serve the crucial function of identifying the range of states attainable by a hybrid system starting from a specified initial state. Demonstrating efficacy in analyzing the behavior of hybrid systems, reachability analysis methodologies also play a pivotal role in the formulation of tailored control strategies for such systems [139,140].

## Data Availability

Not applicable.

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
