# Peer review of "Survey of Advanced Nonlinear Control Strategies for UAVs: Integration of Sensors and Hybrid Techniques"

_sensors, 2024, doi:10.3390/s24113286_

Round 1

Reviewer 1 Report

Comments and Suggestions for Authors

Based on nonlinear control strategies for UAVs, the authors summarized integrating sensors and hybrid techniques. The idea of this paper is good. However, I feel the content of this paper is insufficient. For example, the theme of the paper is UAVs’ nonlinear control strategies, there is no overview of dynamics and control models in the paper, it is difficult to embody the important of nonlinear control strategies. It is recommended the authors supplement relative contents.

Comments on the Quality of English Language

Moderate editing of English language required.

Author Response

Response to Reviewer 1:

We appreciate for confirming our thesis contribution for revision. Your insightful comment regarding the important key point that required modifications has been incredibly valuable to us. We are pleased to inform you that we have taken your comment seriously and have thoroughly addressed it in the revised thesis. Your feedback has contributed significantly to enhancing the quality and rigor of our research work, and we are confident that the revisions made will further strengthen the content of the thesis. A detailed response mentioning comment changes is presented below for your kind consideration:

Comment 1: Based on nonlinear control strategies for UAVs, the authors summarized integrating sensors and hybrid techniques. The idea of this paper is good. However, I feel the content of this paper is insufficient.

Answer: Thank you for your constructive feedback on our review article focusing on nonlinear control strategies for UAVs. We appreciate your acknowledgment of the paper's main idea.

We have considered your comments and expanded the content to provide a more comprehensive overview. Specifically, we have delved deeper into the integration of sensors and hybrid control techniques, elaborating on their significance and practical applications in UAV operations.

We believe that these revisions address the concerns regarding the content's sufficiency and enhance the paper's overall quality and depth. We hope you find the updated version more comprehensive and insightful.

Thank you once again for your valuable feedback, which has helped us improve the quality of our review article.

Modifications: Page 14, Line 397-425

Modifications: Page # 25, Line 856-870

Modifications: Page # 43, Line 1694-1710

Comment 2: For example, the theme of the paper is UAVs’ nonlinear control strategies, there is no overview of dynamics and control models in the paper, and it is difficult to embody the importance of nonlinear control strategies. It is recommended the authors supplement relative contents.

Answer: Thank you for your constructive feedback. We have modified the review article to improve clarity and innovation. We have added the dynamical model of the TRMS in the modified version which is also highlighted in the revised version. We also revise the content of the manuscript to make it more clear.  The conclusion is revised to summarize the innovation.

Modifications: Page 23 & 24, Line 792-811

Modifications: Page 22, Line 728-747

Reviewer 2 Report

Comments and Suggestions for Authors

Introduction and Background Sections: Condense the historical overview of UAV control strategies to highlight key advancements directly leading to the current research focus. Take a few references into consideration for review and cite them: 

a) doi: 10.1109/TIV.2024.3352613

b) doi.org/10.1016/j.geits.2023.100130

Section on Nonlinear Control Strategies: This section should clearly distinguish between the different nonlinear control strategies discussed and their unique advantages for UAV applications.

Sensor Integration in UAVs: Include recent examples of sensor integration that have led to significant improvements in UAV control systems to illustrate the practical impact.

c) doi.org/10.1002/ett.4916

Challenges in UAV Control: Elaborate on specific, contemporary challenges in UAV control not addressed by existing methodologies, setting the stage for the significance of the paper’s review.

Author Response

Response to Reviewer 2:

We appreciate for confirming our thesis contribution for revision. Your insightful comment regarding the important key point that required modifications has been incredibly valuable to us. We are pleased to inform you that we have taken your comment seriously and have thoroughly addressed it in the revised thesis. Your feedback has contributed significantly to enhancing the quality and rigor of our research work, and we are confident that the revisions made will further strengthen the content of the thesis. A detailed response mentioning comment changes is presented below for your kind consideration:

Comment 1: Introduction and Background Sections: Condense the historical overview of UAV control strategies to highlight key advancements directly leading to the current research focus. Consider a few references for review and cite them: 

  1. doi: 10.1109/TIV.2024.3352613
  2. org/10.1016/j.geits.2023.100130

Answer: Thank you for your constructive feedback on our review article focusing on nonlinear control strategies for UAVs. We appreciate your acknowledgment of the paper's main idea.

We have considered your comments and expanded the recent references to compare with the novelty of the proposed review. References are added and highlighted in the revised version.

Modifications: Page # 1, Line 21,23

Comment 2: Section on Nonlinear Control Strategies: This section should clearly distinguish between the different nonlinear control strategies discussed and their unique advantages for UAV applications.

Answer: Thank you for your valuable feedback on our review article. We appreciate your suggestion to clarify the distinction between the different nonlinear control strategies discussed and their unique advantages for UAV applications. We recognize the importance of providing a clear and comprehensive overview of these strategies to assist readers in understanding their applicability and benefits.

In response to your comment, we have revised the "Nonlinear Control Strategies" section to highlight the distinct characteristics and advantages of each strategy, specifically tailored to UAV applications. We believe that these enhancements will address your concern and provide readers with a clearer understanding of the nonlinear control strategies presented in our article.

We hope that our revisions meet your expectations, and we thank you once again for your constructive feedback.

Modifications: Page # 25, Line 856-870

Comment 3: Sensor Integration in UAVs: Include recent examples of sensor integration that have led to significant improvements in UAV control systems to illustrate the practical impact.

Answer: Thank you for your valuable feedback on our review article. We have added a subsection 10.1. Significance of Sensor Integration in UAVs’’ to elaborate its practical impact and also add subsection “8.1. Real Word Applications’’. After that recent references are also added to improve the review article. All modifications are highlighted in the revised version.

Modifications: Page # 43, Line 1694-1710

Modifications: Page # 45, Line 1802-1818

The references are included in the revised version according to your suggestion. References can be checked in the revised version at [1], [2], and [208].

Reviewer 3 Report

Comments and Suggestions for Authors
  • The Abstract and Conclusion sections exhibit significant redundancy, with no new information presented in the Conclusion that was not already covered in the Abstract. It is advisable to delineate clear distinctions between these sections, with the Conclusion adding evaluative comments or insights beyond what is presented in the Abstract.
  • While the paper introduces several advanced control strategies, there is a noticeable absence of mathematical formulations for most, except for Sliding Mode Control (SMC). It is crucial to provide these formulations to substantiate the strategies discussed, particularly when addressing the control of UAVs.
  • The organization of the paper does not facilitate a coherent understanding of the interconnectedness of the topics discussed. This is particularly evident in sections like “Open Research Questions and Future Research Directions,” where the presentation resembles a list rather than a well-integrated narrative. Enhancing the logical flow between sections would greatly improve readability and comprehension.
Comments on the Quality of English Language

No Comment.

Author Response

Response to Reviewer 3:

We appreciate for confirming our thesis contribution for revision. Your insightful comment regarding the important key point that required modifications has been incredibly valuable to us. We are pleased to inform you that we have taken your comment seriously and have thoroughly addressed it in the revised thesis. Your feedback has contributed significantly to enhancing the quality and rigor of our research work, and we are confident that the revisions made will further strengthen the content of the thesis. A detailed response mentioning comment changes is presented below for your kind consideration:

Comment 1: The Abstract and Conclusion sections exhibit significant redundancy, with no new information presented in the Conclusion that was not already covered in the Abstract. It is advisable to delineate clear distinctions between these sections, with the Conclusion adding evaluative comments or insights beyond what is presented in the Abstract.

Answer: Thank you for your constructive feedback on our review article focusing abstract and conclusion. According to the suggestion, the conclusion is revised to make it more comprehensive and reader-friendly. Modifications are also highlighted in the revised version.

Modifications: Page 45 & 46, Line 1835-1863

Comment 2: While the paper introduces several advanced control strategies, there is a noticeable absence of mathematical formulations for most, except for Sliding Mode Control (SMC). It is crucial to provide these formulations to substantiate the strategies discussed, particularly when addressing the control of UAVs.

Answer: Thank you for your constructive feedback. The mathematical formulation of a few basic control strategies is included in the revised version according to the suggestion. Modifications are also highlighted in the revised version.

Modifications: Page 23 & 24, Line 792-811

Modifications: Page 22, Line 728-747

Modifications: Page 20 & 21, Line 649-666

Comment 3: The organization of the paper does not facilitate a coherent understanding of the interconnectedness of the topics discussed. This is particularly evident in sections like “Open Research Questions and Future Research Directions,” where the presentation resembles a list rather than a well-integrated narrative. Enhancing the logical flow between sections would greatly improve readability and comprehension.

Answer: Thank you for your constructive feedback. The paper's organization is revised to make it clearer and a few subsections are added. The modified organization of the article will be more integrated regarding narrative and logical for readers. All modifications are highlighted in the revised version.

Reviewer 4 Report

Comments and Suggestions for Authors

The manuscript provides a comprehensive overview of advanced nonlinear control strategies for UAVs, incorporating various techniques from sliding mode control to adaptive and hybrid strategy. The depth of coverage on how these strategies integrate with UAV systems, particularly in addressing the challenges of nonlinearity and sensor data complexity, is commendable. Here are some suggestions as following:

1. Consider briefly discussing the evolution of UAV control strategies over time to give readers a historical context. Highlight the unique contributions of your work in comparison to existing literature early on to capture the reader's interest.

2. In the Nonlinear Control Strategies Section, for each control strategy discussed, consider providing a specific example of a UAV application where it has been successfully implemented. This can help readers appreciate the practical implications of your research. Include a subsection on potential drawbacks or challenges of each control strategy to provide a balanced perspective.

3. Ensure all references are current and adequately cited within the text. Consider including recent studies to show the manuscript's relevance to current research trends.

4. If possible, for case studies or real-world applications, incorporate case studies or examples of real-world applications of these control strategies in UAV operations. This can illustrate the practical impact of your research findings.

Comments on the Quality of English Language

1. Improving transitions can help readers follow the progression of ideas more smoothly. Consider adding introductory or concluding remarks to each section to summarize key points and how they relate to the next topic.

2. Some sentences with complex technical information could be broken down into shorter, more straightforward sentences. This could make the manuscript more accessible to readers not profoundly familiar with the specific field of UAV control systems.

Author Response

Response to Reviewer 4:

We appreciate for confirming our thesis contribution for revision. Your insightful comment regarding the important key point that required modifications has been incredibly valuable to us. We are pleased to inform you that we have taken your comment seriously and have thoroughly addressed it in the revised thesis. Your feedback has contributed significantly to enhancing the quality and rigor of our research work, and we are confident that the revisions made will further strengthen the content of the thesis. A detailed response mentioning comment changes is presented below for your kind consideration:

The manuscript provides a comprehensive overview of advanced nonlinear control strategies for UAVs, incorporating various techniques from sliding mode control to adaptive and hybrid strategy. The depth of coverage on how these strategies integrate with UAV systems, particularly in addressing the challenges of nonlinearity and sensor data complexity, is commendable. Here are some suggestions as following:

Comment 1: Consider briefly discussing the evolution of UAV control strategies over time to give readers a historical context. Highlight the unique contributions of your work in comparison to existing literature early on to capture the reader's interest.

Answer: Thank you for your constructive feedback on our review article focusing unique contributions of our work. We have modified the revised version according to the suggestion and modifications are also highlighted. A few references are also added to support the modifications.

Modifications: Page 14, Line 397-425

Comment 2: In the Nonlinear Control Strategies Section, for each control strategy discussed, consider providing a specific example of a UAV application where it has been successfully implemented. This can help readers appreciate the practical implications of your research. Include a subsection on potential drawbacks or challenges of each control strategy to provide a balanced perspective.

Answer: Thank you for your constructive feedback. A few basic control strategies in the Nonlinear Control Strategies Section are modified regarding the mathematical implementation of UAV as an example in the revised version according to the suggestion. We also provide evolution in control strategies for UAV control with examples. We also mentioned the limitations and drawbacks of linear and nonlinear control strategies. Modifications are also highlighted in the revised version.

Modifications: Page #  23 & 24, Line 792-811

Modifications: Page #  22, Line 728-747

Modifications: Page #  20 & 21, Line 649-666

Modifications: Page # 25, Line 856-870

Modifications: Page # 43, Line 1694-1710

Modifications: Page # 45, Line 1802-1818

Comment 3: Ensure all references are current and adequately cited within the text. Consider including recent studies to show the manuscript's relevance to current research trends.

Answer: Thank you for your constructive feedback. References are revised and recent articles are included as recent research trends. Modifications are also highlighted in the revised version.

Comment 4: If possible, for case studies or real-world applications, incorporate case studies or examples of real-world applications of these control strategies in UAV operations. This can illustrate the practical impact of your research findings.

Answer: Thank you for your valuable feedback on our review article. We have added a subsection 10.1. Significance of Sensor Integration in UAVs’’ to elaborate its practical impact and also add subsection “8.1. Real Word Applications’’. After that recent references are also added to improve the review article. All modifications are highlighted in the revised version.

Modifications: Page # 43, Line 1694-1710

Modifications: Page # 45, Line 1802-1818

Comments 5 & 6: Improving transitions can help readers follow the progression of ideas more smoothly. Consider adding introductory or concluding remarks to each section to summarize key points and how they relate to the next topic. Some sentences with complex technical information could be broken down into shorter, more straightforward sentences. This could make the manuscript more accessible to readers not profoundly familiar with the specific field of UAV control systems.

Answer: Thank you for your valuable feedback on our review article. The whole paper is revised to improve English and more meaningful. I hope the modifications are satisfactory.
